# Bridging Efficiency and Safety: Formal Verification of Neural Networks with Early Exits

## Abstract

Ensuring the safety and efficiency of AI systems is a central goal of modern research. Formal verification provides guarantees of neural network robustness, while early exits improve inference efficiency by enabling intermediate predictions. Yet verifying networks with early exits introduces new challenges due to their conditional execution paths. In this work, we define a robustness property tailored to early exit architectures and show how off-the-shelf solvers can be used to assess it. We present a baseline algorithm, enhanced with an early stopping strategy and heuristic optimizations that maintain soundness and completeness. Experiments on multiple benchmarks validate our framework's effectiveness and demonstrate the performance gains of the improved algorithm. Alongside the natural inference acceleration provided by early exits, we show that they also enhance verifiability, enabling more queries to be solved in less time compared to standard networks. Together with a robustness analysis, we show how these metrics can help users navigate the inherent trade-off between accuracy and efficiency.

## 1 Introduction

Deep Neural Networks (DNNs) are increasingly deployed in critical domains such as virtual assistants (Gulati et al.) and medical diagnostics (Huang et al., 2023), making their reliability essential. Yet, they are vulnerable to adversarial perturbations: small input modifications that can cause incorrect predictions (Szegedy et al.). This vulnerability has driven extensive research on adversarial attacks and defenses (Costa et al., 2024), highlighting the need for robust and trustworthy AI systems.

Formal verification has emerged as an effective approach for ensuring DNN correctness with respect to specified properties (Katz et al., 2017; Ehlers, 2017; Tjeng et al., 2019; Wang et al., 2021). It rigorously analyzes a network's behavior to guarantee compliance with critical requirements across all possible inputs within a defined domain (Liu et al., 2021). By providing mathematical guarantees for properties like robustness and safety, it offers a valuable tool for building reliable AI systems and supports adoption in high-stakes domains where reliability is crucial (Dalrymple et al.; Russell, 2022).

In addition to robustness and safety issues, another limitation of DNNs lies in their high computational cost, which makes both training and inference power consuming (Elhoushi et al., 2024; Tang et al., 2023; Wright et al., 2024) and limits their use in low-resource systems (Rongkang Dong, 2022; Dimitriou et al., 2024; Ayyat et al., 2024). Even for relatively simple inputs, the inference process of a DNN can be unnecessarily complex and time-consuming. A promising avenue for addressing this computational burden is the use of dynamic inference techniques, such as early exit (EE) (Teerapittayanon et al., 2016; Wang et al., a). EE mechanisms allow a network to terminate computation prematurely once a sufficiently confident prediction is reached at an intermediate stage, thereby reducing computational overhead without compromising accuracy. EE has been adopted in a wide range of domains, including NLP (Elhoushi et al., 2024), Vision (Tang et al., 2023), and speech recognition (Wright et al., 2024), and is increasingly recognized as a powerful tool for optimizing DNN performance in resource-constrained environments (Rongkang Dong, 2022; Yang et al., 2024; Rahmath P et al., 2024; Samikwa et al., 2022; Zhang et al., 2025).

Although EE strategies have demonstrated their potential to enhance runtime efficiency, their implications for formal verification remain largely unexplored. The architectural modification of adding

intermediate exits introduces two key challenges. First, the execution flow can vary, posing technical difficulties for classical verification techniques that assume a fixed output layer. Second, the verification of conditional decision logic must be adapted accordingly.

We address this gap by introducing the formal verification of DNNs with EEs. Our focus is on local robustness, a property that ensures the network's predictions remain consistent within a small neighborhood around a given input. To this end, we propose an algorithm tailored to verify local robustness in DNNs with early exits, enhanced with heuristics that effectively reuse partial results to minimize redundancy and improve scalability. These advances provide a robust framework for verifying DNNs with EEs, contributing to both their reliability and their practical usability in real-world applications. We further leverage our algorithm to enable *early verification* of standard networks by augmenting them with early exits.

In this work, we contribute to the formal verification of DNNs with EEs by: (i) formalizing robustness queries for such networks; (ii) proposing a general algorithm along with two improvements - one checks for early stopping within the verification loop, the other applies heuristics to reduce sub-queries; (iii) leveraging our technique to advance the verification of other models; (iv) analyze when and why the complexity of our algorithm will be smaller than the complexity of standard verification queries; (v) conducting extensive experiments demonstrating the method's practicality and the role of EEs in improving verifiability, and (vi) analyzing how EE training affects verification time, including the impact of thresholds and early stopping.

## 2 PRELIMINARIES

### 2.1 NOTATIONS

A DNN is represented as a function $\mathcal{N} : \mathbb{R}^n \rightarrow \mathbb{R}^m$, where $n, m \in \mathbb{N}$ are input and output dimensions, respectively. For an input $\mathbf{x} \in \mathbb{R}^n$, $\mathcal{N}(\mathbf{x})$ outputs a vector $\mathbf{y} \in \mathbb{R}^m$. We focus on classification networks, where the predicted class is the index of the highest value in $\mathbf{y}$ (the *winner*); other indices are *runner-ups*. An $\epsilon$-ball around $x$, denoted $B_\epsilon^x$, is the set $\{x' \in \mathbb{R}^n : \|x' - x\| \leq \epsilon\}$.

### 2.2 FORMAL VERIFICATION OF DNNS

The formal verification of a DNN $\mathcal{N} : \mathbb{R}^n \rightarrow \mathbb{R}^m$ can be cast into a constraint satisfiability problem, where the goal is to determine whether a property $P$ is *satisfiable* in $\mathcal{N}$. $P$ represents the existence of an input to $\mathcal{N}$ within a specific domain $\mathcal{D}$ whose output satisfies a constraint $\phi$:

$$\exists \mathbf{x} \in \mathcal{D} \text{ such that } \phi(\mathbf{x}, \mathcal{N}(\mathbf{x})) \text{ is satisfied.}$$

If $P$ is satisfiable, $\mathcal{N}$ is said to be UNSAFE with respect to $\phi$. Typically, $\phi$ captures undesirable behavior by encoding the negation of a desired property. If $P$ is not satisfiable, the desired property holds and $\mathcal{N}$ is SAFE.

### 2.3 DNNs WITH EARLY EXITS

A DNN with early exits is a network augmented with additional decision points, known as *exits*, within its architecture. These exits allow the network to terminate inference early if a condition, typically a confidence threshold, is met, reducing computational cost while maintaining accuracy. Let $\mathcal{N}_{ee} : \mathbb{R}^n \rightarrow \mathbb{R}^m$ denote a network with $k$ exits, and let $\mathbf{y}^{(j)}$ represent the neuron values of the $j$-th exit. Inference at exit $j$ terminates if:

$$f(\mathbf{y}^{(j)}) \geq T_j,$$

where $T_j$ is a predetermined confidence threshold for the $j$-th exit, and $f$ is a confidence function used to evaluate whether the exit condition is satisfied. In early works (Teerapittayanon et al., 2016), $f(\mathbf{y}^{(j)})$ was computed as the entropy of the $j$'th exit logits; and later, alternative gating mechanisms were proposed to improve the efficiency and accuracy of EE mechanisms. These include using the maximum *SoftMax* probability as a confidence measure (Panda et al., 2016), leveraging confidence accumulation across multiple layers (Scardapane et al., 2020), and dynamically learning the optimal exit conditions (Xin et al., 2021). Regardless of the specific gating function $f$, the final output $\mathbf{y}$

is determined by the first exit where the condition $f(\mathbf{y}^{(j)})$ is satisfied. If no such condition is met, the output is taken from the last exit. In this work, we use a fully connected layer followed by a *SoftMax* activation as the confidence function $f$, producing $\mathbf{y}^{(j)}$, and adopt the straightforward threshold condition: $\max(\mathbf{y}^{(j)}) > T_j$. A common threshold value, which we also use in many of our experiments, is $T = 0.9$ (Rahmath P et al., 2024; Rongkang Dong, 2022); although we also experimented with other values, all greater than $0.5$. (Setting $T$ to values lower than $0.5$ can result in multiple classes exceeding the threshold, and we ignore such cases). Fig. 7 in App. A depicts a DNN with EEs.

## 3 VERIFYING DNNs WITH EARLY EXITS

In the context of DNN verification, networks with EEs present both opportunities and challenges. On one hand, the inference process in such networks often concludes in earlier layers, potentially reducing the size of the network that needs to be verified. On the other hand, adding EEs introduces two major complexities to the traditional formal verification of DNNs:

1. In conventional DNNs, the output property is defined on a single output layer. For networks with EEs, this definition must be adapted to accommodate multiple output layers.

2. Second, the presence of multiple output layers and the inference logic introduces conditional branching: If the current exit yields a confident prediction, it returns the result; else, computation proceeds to the next layer. This conditional behavior must be incorporated into the verification process to avoid spurious counterexamples where the runner-up wins at exit $e$ while the winner prevails at an earlier exit $e' < e$.

Here, we propose a general framework for verifying DNNs with EEs, addressing the challenges outlined above. We begin in 3.1 by redefining local robustness to accommodate multiple exits. Then, in 3.2, we present a basic verification algorithm that mirrors the conditional inference process of EE. In 3.3, we analyze the complexity of this approach and show that, under certain conditions, its cost can be significantly reduced. Finally, in 3.4, we suggest an improved algorithm that incorporates two key optimizations to reduce redundant queries while preserving soundness and completeness.

### 3.1 REVISED ROBUSTNESS PROPERTY

For a standard DNN $\mathcal{N}$ (without EEs), the property to negate the local robustness of $\mathcal{N}$ around a sample $x$ is typically defined as:

$$P := \exists x' \in B_\epsilon^x, \exists i \in \mathcal{C} \text{ such that } \mathcal{N}(x')_i > \mathcal{N}(x')_w$$

where $\mathcal{C}$ is the set of possible output labels $\{1, \ldots, m\}$, $B_\epsilon^x$ is the $\epsilon$-ball around $x$ (plays the role of $\mathcal{D}$ in the definition), $w$ is the index of the winner class and $\mathcal{N}(x)_j$ is the $j$'th value in $\mathcal{N}(x)$.

In DNNs with EEs, the inference process enables outputs to be returned from various exits in the network, corresponding to different neurons. This adds ambiguity to the traditional definition, as it is unclear which exit represents the output of $\mathcal{N}(x)$, with all exits being potential candidates. The verification process must therefore condition the validity of the specification on the assumption that a specific exit serves as the actual output layer for the given input.

A counterexample to robustness of a network with EEs is one where (i) a "runner-up output" wins in an early or output exit $e$, and (ii) the "true", desired output does not prevail at any exit preceding $e$. These conditions are encapsulated in the following property $P_{ee}$, which defines the negation of the revised local robustness property for a DNN with EEs. The indices of the layers with EEs and the index of the output layer are denoted with $ee$ and $last$, respectively.

$$P_{ee} := \exists x' \in B_\epsilon^x, i \in \mathcal{C} \setminus \{w\}, e \in ee \cup \{last\} :$$
$$((\mathcal{N}(x')_i^e > T_e \wedge e < last) \quad \vee \quad (\mathcal{N}(x')_i > \mathcal{N}(x')_w \wedge e = last)) \quad \wedge$$
$$\forall j \in ee \cap \{1, \ldots, e-1\} : \mathcal{N}(x')_w^j < T_j$$

Here, $P_{ee}$ asserts that (first line) there exists an input $x'$, a runner-up $i$, and an exit $e$ such that (second line) the runner-up wins in the early exit (left side) or in the last layer (right side), and (third line) the winner does not prevail at any earlier exit. If $P_{ee}$ is satisfiable, the network is UNSAFE to be robust within an $\epsilon$-ball around $x$. Otherwise, its negation is valid and the network is SAFE.

## 3.2 VERIFICATION FRAMEWORK

To verify robustness in DNNs with EEs, we propose Alg. 1. The algorithm operates by iterating through all exits (outer loop). For each exit, it examines each runner-up class (inner loop) to determine whether there exists a counterexample where the runner-up wins, and the output is produced at the current exit. This is accomplished by a verification query (line 5 or 7) that tries to satisfy the following property: the runner-up wins, and the winner has not already won in any preceding exit. If UNSAFE is returned, the resulting example satisfies $P_{ee}$, thereby providing a counterexample to the robustness of $\mathcal{N}$. Otherwise, if SAFE is returned, the local robustness of $\mathcal{N}$ is verified. Note that the call to Verify in line 9 launches an underlying verification tool to solve a standard verification query.

---

**Algorithm 1** Verify Local Robustness in DNNs With Early Exits

---

**Input** $\mathcal{N}$, $x$, $\epsilon$ **Output** $\mathcal{N}$ is robust in $B_\epsilon^x$, or counterexample
1: $w, ee, last = argmax(\mathcal{N}(x))$, indices of layers with EEs, index of $\mathcal{N}$'s last layer
2: **for** $k \in ee \cup \{last\}$ **do**
3:     **for** $i \in \mathcal{C} \setminus \{w\}$ **do**
4:         **if** k $\neq$ last **then**
5:             $\mathcal{P} := \exists x' \in B_\epsilon^x : (\mathcal{N}(x')_i^k > T_k) \wedge (\forall e \in ee \cap \{1, \dots, k-1\} : \mathcal{N}(x')_w^e < T_e)$
6:         **else**
7:             $\mathcal{P} := \exists x' \in B_\epsilon^x : (\mathcal{N}(x')_w < \mathcal{N}(x')_i) \wedge (\forall e \in ee : \mathcal{N}(x')_w^e < T_e)$
8:         **end if**
9:         res, cex = Verify($\mathcal{N}$, $B_\epsilon^x$, $\mathcal{P}$)
10:         **if** res == UNSAFE **then**
11:             **return** UNSAFE, cex
12:         **end if**
13:     **end for**
14: **end for**
15: **return** SAFE

---

**Theorem 1.** *If the underlying verifier is sound and complete, Alg. 1 is sound and complete.*

## 3.3 FIXED PARAMETER TRACTABLE COMPLEXITY

Alg. 1 contains two nested loops; while it returns immediately upon finding a counterexample (UNSAFE), it must exhaust the entire loop before concluding SAFE. This motivates further improvements, as we show that in some cases, local robustness in DNNs with early exits can be determined more efficiently. For that purpose, we define the *trace* of an input and its stability as follows.

**Definition 3.1.** The trace $\tau(x)$ of an input $x$ in DNN $\mathcal{N}$ with EEs is the set of layers $x$ is propagated through. Given an $\epsilon > 0$, $\tau(x)$ is *stable* in $B_\epsilon^x$ if $\forall x' \in B_\epsilon^x : \tau(x') = \tau(x)$.

The trace of an input determines which parts of the network must be considered to verify robustness for that input. Suppose there exist $x$ and $\epsilon > 0$ such that all $x' \in B_\epsilon^x$ share the same trace as $x$. Then, if Alg. 1 does not return UNSAFE before reaching the exit layer of $x$, it will eventually return SAFE, as no more paths can be checked. Hence, under the trace stability assumption, the complexity of solving $P_{ee}$ depends on $|\tau(x)|$, the number of layers in $\tau(x)$, rather than the total number of layers in $\mathcal{N}$; any iterations beyond that point are redundant. In Sec. 4 (Fig. 3), we show that the trace stability holds in practice. To analyze the complexity, we remind the reader the definition of a *Fixed Parameter Tractable* (*FPT*) problem, and, focusing in ReLU networks, use it to express the complexity of the verification.

**Definition 3.2** ((Downey, 2012, Def. 1)). A problem is *Fixed-Parameter Tractable* (*FPT*) with respect to a parameter $p$, (denoted as $FPT(p)$), if it can be solved in time $f(p) \cdot poly(n)$, where $f$ is a computable function of $p$, and $n$ is the input size.

We use this class to show that there are cases in which only a partial part of the network can be considered, and not all the network, leading to much better worst case scenario complexity.

**Theorem 2.** *Given a network $\mathcal{N}$ with EEs and ReLU activations, layer width bound $k$, input $x$, and $\epsilon > 0$, if $\tau(x)$ is stable in $B_\epsilon^x$, then solving $P_{ee}$ with $(\mathcal{N}, x, \epsilon)$ is $FPT(k \cdot |\tau(x)|)$.*

In the following subsection, we try to improve Alg. 1 to have $FPT(k \cdot |\tau(x)|)$ complexity under the assumptions above and also to save additional redundant queries.

### 3.4 ADDITIONAL OPTIMIZATIONS

As noted, proving SAFE with Alg. 1 requires calling Verify (line 9) for every exit and every class. As it makes the process time consuming, we discuss two improvements to expedite the overall procedure. To maintain readability and conciseness, we denote lines 3-13 in the algorithm as the function $ExistsPrevCEX(\mathcal{N}, x, \epsilon, k, T, last, ee, w)$ and, more compactly, as $ExistsPrevCEX(\mathcal{N}, x, \epsilon, k)$. These lines rule out adversarial examples in earlier layers.

One potential improvement involves reducing the number of queries for all classes at each exit layer - thereby avoiding the inner loop on line 3 of Alg. 1, and *continuing* to the next iteration of the loop on line 2. Instead of verifying that the confidence of every class is below the threshold, the modified algorithm initially checks whether the winner's score is greater than $1 - T$. This condition ensures that the score for no other class can exceed $T$. If this check fails, the algorithm must fall back to verifying each class individually; but we empirically observed that often this condition is met, and the inner loop can be skipped. This optimization is implemented in Alg. 2 (orange lines).

Alg. 2 further improves Alg. 1 by adding a mechanism to determine robustness earlier, without exhaustively exploring all possible runner-up labels in all exits. Specifically, at each exit layer, it checks (blue lines) whether the winner's score always exceeds the threshold. If this condition holds and earlier iterations have confirmed no counterexamples exist in prior exits, it guarantees that all inputs advance to the current exit, where the original winner consistently prevails. In such scenarios, the algorithm can *break* the iteration on the loop on line 2 at Alg. 1 and soundly return SAFE without further checks in next exits.

---

**Algorithm 2** Verify DNNs with Early Exits - *Break then Continue* Optimizations

**Input** $\mathcal{N}, x, \epsilon_p$ **Output** $\mathcal{N}$ is robust in $B_\epsilon^x$, or counterexample
1: $w, ee, last = argmax(\mathcal{N}(x))$, indices of layers with EEs, index of $\mathcal{N}$'s last layer
2: **for** $k \in ee \cup \{last\}$ **do**
3:     **if** k $\neq$ last **then**
4:         $\mathcal{P} := \exists x' \in B_\epsilon^x : \mathcal{N}(x')_w^k < T$
5:     **else**
6:         $\mathcal{P} := \exists x' \in B_\epsilon^x, \exists i \in \mathcal{C} \setminus \{w\} : \mathcal{N}(x')_w < \mathcal{N}(x')_i$
7:     **end if**
8:     res, cex = Verify($\mathcal{N}, B_\epsilon^x, \mathcal{P}$)
9:     **if** res == SAFE **then**
10:       **return** SAFE
11:     **end if**
12:     $res, cex$ = Verify($\mathcal{N}, B_\epsilon^x, \forall x' \in B_\epsilon^x : \mathcal{N}_w^k(x') < 1 - T$)
13:     **if** k == last $\vee$ res == UNSAFE **then**
14:       res, cex = $ExistsPrevCEX(\mathcal{N}, x, \epsilon, k)$
15:       **if** res == UNSAFE **then**
16:         **return** UNSAFE, cex
17:       **end if**
18:     **end if**
19: **end for**
20: **return** SAFE

---

**Theorem 3.** *If the underlying verification tool is sound and complete, Alg. 2 is sound and complete.*

**Theorem 4.** *Given a network $\mathcal{N}$ with EEs and ReLU activations, layer width bound $k$, input $x$, and $\epsilon > 0$, if $\tau(x)$ is stable in $B_\epsilon^x$, then Alg. 2 runtime is $\mathcal{O}(2^{k \cdot |\tau(x)|}) \cdot poly(\#neurons\ in\ \mathcal{N})$.*

To summarize this section, we formalized a robustness property for DNNs with EEs and proved it is fixed-parameter tractable (Thm. 2) under trace-stability assumption. We presented a sound and complete baseline algorithm (Alg. 1), then introduced break-and-continue heuristics (Alg. 2) and showed they yield a tighter complexity bound (Thm. 4). Proofs of algorithm soundness and completeness (Thms. 1 and 3) and of complexity analysis (Thms. 2 and 4) can be found in App. C.

## 4 EVALUATION

We evaluate our method across a diverse set of networks and datasets, focusing on several key aspects. First, we demonstrate the practicality and limitations of our approach across different architectures. Next, we analyze early prediction and early verification behaviors to uncover meaningful insights. Finally, we present ablation studies for each metric to support our conclusions and clarify the impact of our contributions.

### 4.1 EXPERIMENTAL SETUP

We implemented our framework with PyTorch, and ran the experiments on a machine with macOS, 16 GB RAM, and an Apple M3 chip with an 8-core GPU. We further confirmed our results under an additional experimental setting, as shown in App. H. As an underlying verification tool, our framework uses (but is not limited to) Alpha-Beta CROWN (Xu et al., 2020; 2021; Wang et al., 2021), a state-of-the-art method for formally verifying adversarial robustness properties of DNNs. Alpha-Beta CROWN does not currently support the encoding of nested AND operators, needed for our queries. We circumvent this by encoding each query as a collection of smaller queries, one for each conjunct. Then, to enable a fair comparison, we used a similar encoding also for the original queries, even though such an encoding may not be optimal. We note that there does not seem to be any conceptual issue with supporting nested ANDs (e.g., Marabou (Wu et al., 2024)); and once such support is added, our encoding could be simplified.

### 4.2 BENCHMARKS AND MODEL TRAINING

To evaluate the effectiveness of our method, we conducted experiments on several widely recognized datasets: MNIST (LeCun et al., 2010), CIFAR-10 (Krizhevsky, 2009) and CIFAR-100 (Krizhevsky, 2009), with multiple common architectures: Fully Connected, CNN (LeCun et al., 1998), ResNet (He et al., 2016) and VGG (Simonyan & Zisserman, 2015). These were chosen to ensure a diverse and representative assessment of our approach, covering various data complexities, neural architectures, and classification challenges. The full details on the datasets and models used are given in App. D.

We adopt the training procedure from prior work (Teerapittayanon et al., 2016; Zhou et al.). A baseline model is first trained without exits. Then, EEs - each a fully connected layer with *SoftMax* - are added and trained sequentially, keeping the main model fixed. Each exit is optimized individually and frozen before proceeding. Full details are provided in App. D.

### 4.3 EVALUATING THE PRACTICALITY OF VERIFYING EARLY EXIT NETWORKS

Fig. 1 presents the result distribution for our algorithm across various epsilon values and samples. We used 100 examples per benchmark, with fine-grained epsilon values in the range $\epsilon \in \{0.1, 0.05, 0.01, 0.005, 0.001\}$. Each column sums the number of SAFE, UNSAFE and UNKNOWN results for a given epsilon value. Note that UNKNOWN results typically from timeouts (30 minutes per example) or assertion failures in the underlying verifier, often due to loose bounds reflecting the query's complexity, and are not directly produced by our algorithm.

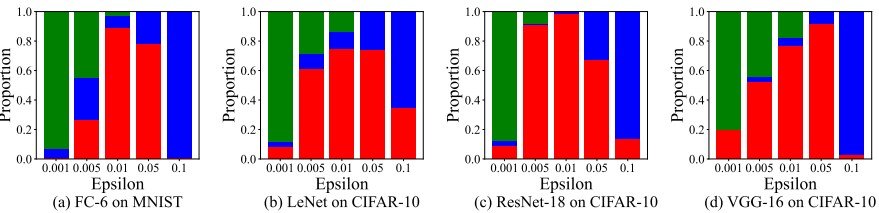

(a) FC-6 on MNIST    (b) LeNet on CIFAR-10    (c) ResNet-18 on CIFAR-10    (d) VGG-16 on CIFAR-10

Figure 1: SAFE/UNSAFE/UNKNOWN counts per epsilon, across different networks and datasets.

The graphs highlight the diversity of local robustness queries (five columns each) and the challenging regions, specifically $\epsilon$ values near the boundary between SAFE and UNSAFE outcomes. The eval-

uation approximates the smallest $\epsilon$ where robustness is quickly verified and the largest where it is quickly refuted, adding additional intermediate $\epsilon$ values in between.

Tab. 1 compares the performance of Alg. 1 and Alg. 2, highlighting the improvements gained by incorporating the *break* and *continue* optimizations. While the differences in UNSAFE cases are minor - since counterexamples, when they exist, are typically found quickly - SAFE cases show a significant improvement, with the optimized algorithm performing up to $10\times$ faster. For a more detailed ablation study of each optimization's contribution, please refer to App. F. Note that in VGG16, Alg. 1 failed to prove SAFE cases before timing out, whereas Alg. 2 succeeded, further showing its effectiveness. In addition to runtime, we report the number of UNKNOWN outcomes (timeouts or inconclusive results), shown in Tab. 2. The optimized algorithm preserves the UNKNOWN rate across most models and reduces it by over 20% in the largest benchmark (VGG16), demonstrating that the improvements are not limited to already-solvable cases.

Table 1: Verification runtime statistics (in seconds) per benchmark. Rows correspond to Alg. 1 (top) and Alg. 2 (bottom).

| Benchmark | UNSAFE | | | SAFE | | |
|---|---|---|---|---|---|---|
| | Mean | Std | Median | Mean | Std | Median |
| MNIST, FC6 | 0.527 | 1.903 | 0.068 | 13.037 | 3.185 | 13.032 |
| | 0.600 | 2.739 | 0.201 | 1.143 | 1.205 | 0.397 |
| CIFAR10, LeNet | 2.250 | 6.670 | 0.320 | 56.927 | 9.516 | 58.628 |
| | 2.279 | 4.556 | 0.688 | 7.255 | 10.550 | 6.185 |
| CIFAR10, ResNet18 | 3.518 | 12.499 | 1.306 | 409.395 | 74.235 | 375.703 |
| | 5.679 | 12.316 | 3.197 | 42.646 | 60.521 | 19.061 |
| CIFAR10, VGG16 | 5.412 | 1.032 | 5.588 | – | – | – |
| | 8.823 | 1.818 | 8.737 | 1532.418 | 348.150 | 1417.474 |

Table 2: Number of UNKNOWN outcomes (lower is better).

| Benchmark | Alg. 1 | Alg. 2 |
|---|---|---|
| MNIST, FC6 | 672 | 652 |
| CIFAR10, LeNet | 253 | 256 |
| CIFAR10, ResNet18 | 159 | 162 |
| CIFAR10, VGG16 | 68 | 53 |

## 4.4 ADDING EARLY EXITS TO STANDARD MODELS

We next compare verification times for networks with and without EEs, to explore the potential of adding EEs and use our technique as a method to improve verifiability of standard models. Fig. 2 illustrates that DNNs with EEs can be verified more efficiently. While verification time for simple queries remains largely unchanged, harder queries are verified significantly faster.

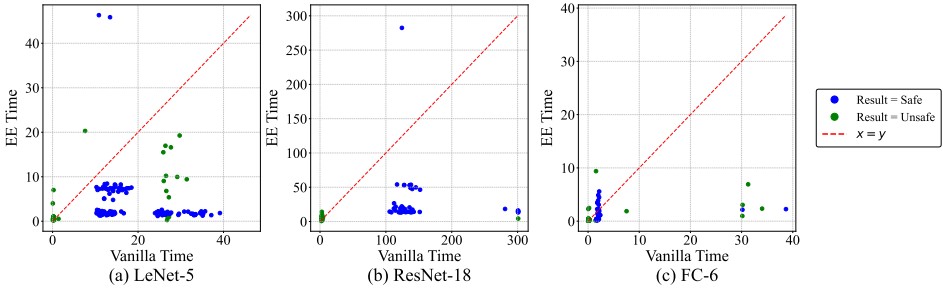

(a) LeNet-5     (b) ResNet-18     (c) FC-6

Figure 2: Comparison of verification times between the original model with the underlying verifier (Vanilla Time) and the model with EEs, using Alg. 2.

We additionally verified a ResNet-18 model on CIFAR-100, a more challenging task due to the larger number of classes. Both the baseline and the model with EEs identified UNSAFE cases. However, the model with EEs verified SAFE examples in about one hour, while the baseline reached a two-hour timeout. For $\epsilon \in \{0.1, 0.01, 0.001\}$ (25 samples each), all samples at $\epsilon = 0.1$ were UNSAFE (with and without EEs); at $\epsilon = 0.01$, most were UNSAFE (20 with EEs and 21 without EEs), with the remainder UNKNOWN; and at $\epsilon = 0.001$, 14 samples were SAFE (verified only with EEs), with the rest UNKNOWN.

To better understand this phenomenon, we compare the exit layers of the inference with those of the verification in Fig. 3. For example, the top-left cell in subfigure (a) indicates that out of 39 samples that were exited in the first exit during ResNet-18 inference, the verification of 37 was improved to exit in the first exit as well, and subfigure (d) indicates that all counterexamples in all UNSAFE

cases where found in the verification of the first exit, independently with the exit of the inference. While no clear correlation is observed for UNSAFE cases (as expected, since the original sample and counterexamples behave differently), a strong correlation emerges for SAFE cases in the first and last exits. This finding suggests that the *trace stability assumption* holds, and the optimization allowing verification to stop earlier is effective, demonstrating that adding EEs can enhance the verifiability of DNNs. Additional results are provided in App. E.

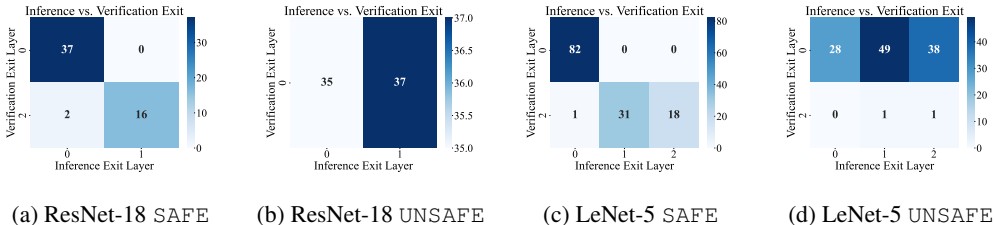

(a) ResNet-18 SAFE  (b) ResNet-18 UNSAFE  (c) LeNet-5 SAFE  (d) LeNet-5 UNSAFE

Figure 3: Heatmaps demonstrating the correlation between the inference and verification exit layers, and the correctness of the *trace stability assumption*.

### 4.5 ROBUSTNESS ANALYSIS

Training networks with EEs entails a trade-off between predictive accuracy and inference latency. Different hyperparameter settings yield distinct working points, and users must select the one that best satisfies their performance or resource constraints.

**Impact of the Early Stop Threshold.** Adjusting the confidence threshold at each exit introduces a trade-off between accuracy and latency. Fig. 4 (left) shows that higher thresholds improve accuracy but also increase inference time. To quantify robustness, we measure it as the proportion of inputs formally verified as SAFE, i.e. #SAFE / (#SAFE + #UNSAFE). With the EE architecture fixed and only the confidence threshold varied, Fig. 4 (middle) shows that robustness closely follows accuracy - higher thresholds boost both metrics. However, Fig. 4 (right) reveals that verification time also grows with the threshold, mirroring the inference-accuracy trade-off. Consequently, selecting a threshold requires balancing several important objectives: accuracy, latency, robustness and verifiability.

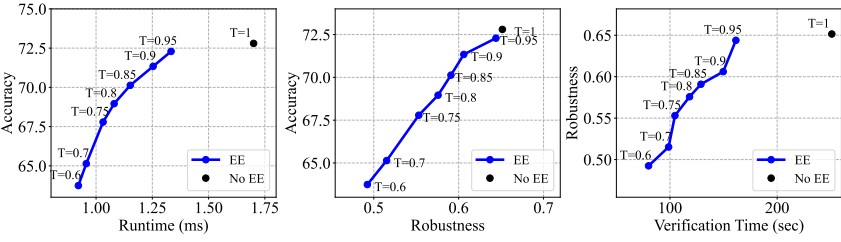

Figure 4: Impact of threshold selection on accuracy vs. runtime (left), accuracy vs. robustness (middle) and robustness vs. verification time (right) for CNN on CIFAR-10, with $\epsilon = 0.005$.

To dissect the verification cost further, Fig. 5 breaks down verification times across several $\epsilon$ values. Both the mean and variance of verification time grow with the threshold, reinforcing that more conservative exit criteria - while safer - demand heavier verification effort. We also compare against the *vanilla* network (without EEs), which is equivalent to threshold $T = 1$. As Fig. 6 shows, the vanilla model achieves the highest robustness but at the cost of the longest verification times - a direct consequence of requiring the full network to be analyzed.

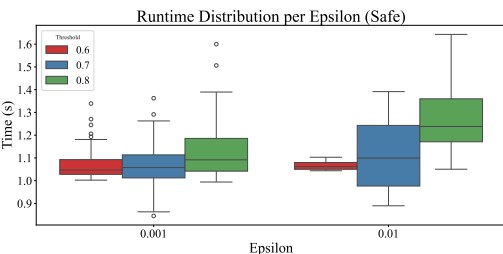
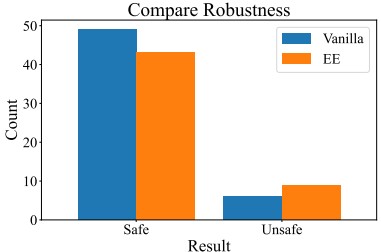

Figure 5: Detailed impact of threshold selection on verification time for LeNet-5 on CIFAR-10.

Figure 6: Robustness comparison between vanilla and models with EEs.

**Impact of the Exit Location.**   In a final set of experiments, we fixed the EE architecture and varied the confidence threshold $T$. Here, we fix $T$ and compare two ResNet-18 variants that differ only in the position of the single exit: one after block 1 (**RN_ee1**) and the other after block 2 (**RN_ee2**). Tab. 3 reports each model's accuracy, average inference latency, and verification outcomes.

Table 3: Verification statistics for ResNet-18 variants with a single early exit.

| Architecture | Accuracy | Inference Time | #SAFE | #UNSAFE | #UNKNOWN | Safe Ver Time | Unsafe Ver Time |
|---|---|---|---|---|---|---|---|
| **RN_ee1** | 0.8757 | 30.56ms | 121 | 109 | 190 | 21.4s | 11.7s |
| **RN_ee2** | 0.8921 | 36.94ms | 85 | 108 | 227 | 23.4s | 14.2s |

We find that placing the exit earlier (after block 1) raises the robustness - the share of inputs formally verified as SAFE - while incurring only a small accuracy drop. By contrast, moving the exit deeper (after block 2) yields a slight accuracy gain but lowers both verifiability (longer verification time, more #UNKNOWN examples) and robustness. This shows that exit placement itself is an effective design: EEs close to the input strengthen formal guarantees, whereas later exits preserve more of the network's full expressivity. These insights can help practitioners choose exit locations that best balance accuracy, latency, robustnes and safety requirements.

## 5   RELATED WORK

This work lies at the intersection of improving DNN efficiency and ensuring robustness. The formal verification of DNNs has received growing attention (Liu et al., 2021; Brix et al., 2023; Brix et al.) due to their increasing use in safety-critical domains. Early efforts primarily focused on verifying fully-connected and convolutional networks (Katz et al., 2017; Huang et al., 2017; Gehr et al., 2018; Bunel et al., 2020), while more recent work has expanded to specialized architectures such as RNNs (Khmelnitsky et al.), LSTMs (Moradkhani et al., 2023), transformers (Shi et al.)  and GNNs (Wu et al., 2022; Ladner et al., 2025; Sälzer & Lange, 2023; Hojny et al., 2024).

Various techniques have been proposed to improve verification scalability, including symbolic propagation, abstract interpretation, abstraction-refinement, adversarial pruning, and certified training (Wang et al., 2018a;b; Gehr et al., 2018; MiG, 2018; Singh et al., 2019; Wang et al., 2021; Elboher et al., 2020; Ashok et al., 2020; Elboher et al., 2022; Xu et al., 2021; Zhang et al., 2022; Zhou et al., 2024; Mueller et al., 2023; Palma et al., 2024; Mao et al., 2023). While these approaches enhance verification efficiency, we focus on establishing a framework for optimized networks.

EE represents a dynamic inference strategy within the broader landscape of DNN optimization methods, which also includes static approaches such as quantization (Cheng et al.), pruning (Frankle & Carbin), knowledge distillation (Phuong & Lampert, 2019), and neural architecture search (Zoph & Le). Other dynamic approaches include selective pruning (Gao et al.; Lin et al., 2017), spatial attention (Li et al., 2021; Wang et al., b), and temporal redundancy reduction (Raviv et al., 2022; Dinai et al., 2024). Models with EEs accelerate inference by allowing the network to terminate computation early for easy inputs (Rahmath P et al., 2024; Bajpai & Hanawal; Dimitriou et al., 2024; Samikwa et al., 2022). Common gating strategies include entropy (Teerapittayanon et al., 2016) and

*SoftMax* thresholds (Seon et al., 2023). In this work, we adopt a threshold on the logits, though our framework supports other mechanisms.

Lastly, among the numerous methods that modify the training process to promote formal guarantees (Madry et al., 2018; Dvijotham et al.; Guo et al., 2021; Jin et al., 2022; Chen et al., 2022; Zeqiri et al., 2023), some approaches that aim to make networks easier to verify, incorporate regularization or architectural constraints (Xiao et al., 2019; Xu et al., 2024; Zhang et al., 2023; Shriver et al.; Liu et al., 2025). However, our work is, to our knowledge, the first to explore how EEs themselves can support scalable verification. Concurrently, work on neural activation patterns (NAPs) reveals a related phenomenon: robust behavior in DNNs often depends on only a small subset of neurons (Geng et al., 2023; 2025a;b). These minimal NAPs serve as compact specifications, and more recently, as interpretable logical structures. This aligns with our observation that early exits can certify predictions using only a shallow prefix of the network. Conceptually, NAPs can be seen as a *finer-grained analogue* of early exits, where selected neuron subsets function as micro-exits carrying sufficient evidence.

## 6 FUTURE WORK AND CONCLUSION

**Future Work.**   Our work leaves several opportunities for future research. First, extending the verification framework to encompass other properties, such as safety and fairness, would broaden its applicability. Second, using distributed computing to parallellize the verification and enhance the scalability of our method, particularly for networks with numerous exit points. Third, we assume the basic condition of $\max(\mathbf{y}^{(j)}) > T_j$. Additional exit condition functions can be explored, and more strategies for dynamic inference could be examined (Rahmath P et al., 2024).

**Conclusion.**   Our work lays the groundwork for the verification of DNNs with EEs, aiming to bridge the gap between inference optimization and formal verification. By extending the scope of properties, tools, and methods, future research can continue to advance the reliability and applicability of these networks across diverse domains.

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

# Appendix

The appendix provides visualizations, complexity analysis, proofs, technical details, and additional experiments that could not be included in the main paper.

## A   AN EXAMPLE OF NEURAL NETWORK WITH EARLY EXITS

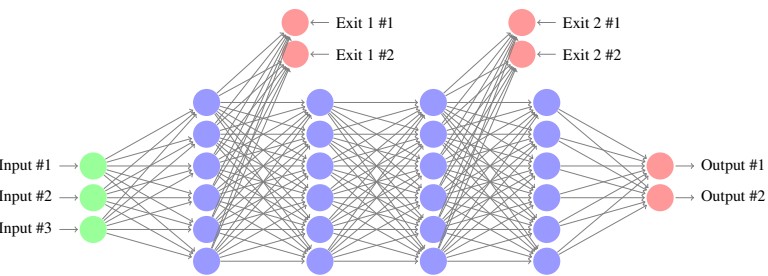

Figure 7: A fully connected DNN with two EEs placed at the first and third layers. The neuron values at these exits are the results of applying *SoftMax* to the hidden values in the corresponding layers.

## B   COMPLEXITY ANALYSIS

### B.1   ALG. 1'S COMPLEXITY

We analyze the complexity of the proposed verification algorithms. We remind that solving verification queries is an NP-Hard problem (Katz et al., 2017), and current methods have exponential complexity in the network's number of neurons in a worst-case scenario. Hence, we denote the worst-case complexity of the underlying verification tool used in our method by $\mathcal{O}(2^N)$ where $N$ is the number of neurons in $\mathcal{N}$.

**Theorem 5.** *The worst-case complexity of Alg. 1 is $\mathcal{O}(2^N)$.*

*Proof.* Alg. 1 applies $E \cdot (C-1) + 1$ verification queries, where $E$ is the number of EEs and $C$ is the number of classes ($+1$ for the last query). Each query is $\mathcal{O}(2^{N_i})$ where $N_i$ is the number of neurons in the partial network until the $i$'th exit (including), resulting in $(E \cdot (C-1) + 1)$ calls to verification problems of $\mathcal{O}(2^m)$ complexity where $m \leq N$, summing to an overall $\mathcal{O}(2^N)$ complexity.     □

### B.2   ALG. 2'S COMPLEXITY

Alg. 4 performs *break* and *continue* heuristic optimizations, which are not necessarily take place, and hence its worst case complexity is similar to that of Alg. 1. However, its complexity decreases under the trace stability assumption. We prove Thm. 4:

**Theorem.** *Given a network $\mathcal{N}$ with EEs and ReLU activations, layer width bound $k$, input $x$, and $\epsilon > 0$, if $\tau(x)$ is stable in $B_\epsilon^x$, then Alg. 2 runtime is $\mathcal{O}(2^{k \cdot |\tau(x)|}) \cdot poly(\#neurons\ in\ \mathcal{N})$.*

*Proof.* We denote the index of the early exit where the output of $x$ is returned by $E_x$. In Alg. 2, line 8 checks if the winner always wins in the current exit. If the trace of each input in $B_\epsilon^x$ is equal to $\tau(x)$ and the result is SAFE, it must be returned in line 8 in the $E_x$'th iteration: it can't be returned before since $x$ itself constitutes a counterexample (as its propagation does not finish before), and it is returned in iteration $E_x$ since the trace of all inputs are equal to $\tau(x)$, and a sound and complete underlying verifier will return SAFE in that case. If the answer is UNSAFE, the counterexample must be found until the $E_x$'th iteration; in each iteration, line 14 launches $ExistsPrevCEX(\mathcal{N}, x, \epsilon, k)$, which checks all possible counterexamples until the $E_x$'th exit. Because $\tau(x') = \tau(x)$ for all inputs, all possible behaviors are examined after reaching line 14 at the $E_x$'th iteration, and the counterexample must be found, given that the underlying verification tool is sound and complete.

This means that Alg. 2 will not proceed to iterations that run verification queries on partial networks bigger than the partial network that $x$ was propagated through. The number of neurons in each partial network is no more than $k \cdot |\tau(x)|$, and solving verification queries with that size can be done in $2^{k \cdot |\tau(x)|} \cdot poly(\text{\#neurons in } \mathcal{N})$, since there are $2^{k \cdot |\tau(x)|}$ options to the activation state - active ($y = x$) or inactive ($y = 0$) - and in each option, solving the linear constraints is $poly(\text{\#neurons in } \mathcal{N})$ (Khachiyan, 1980). Therefore, the complexity of each verification query is not greater than $2^{k \cdot |\tau(x)|} \cdot poly(\text{\#neurons in } \mathcal{N})$. As a result, the running time of Alg. 2 is bounded by $\mathcal{O}(2^{k \cdot |\tau(x)|}) \cdot poly(\text{\#neurons in } \mathcal{N})$. $\square$

Lastly, we mention that using our training method, the trace of inputs in the network with EEs is significantly smaller than in the same network without exits, with high probability (Teerapittayanon et al., 2016), significantly decreasing complexity from the number of neurons in the network to a much smaller number.

## C  PROOFS

In this appendix we provide the detailed proofs for the algorithms along the paper.

### C.1  PROOF OF THEOREM 1

**Theorem.** *If the underlying verification tool is sound and complete, Alg. 1 is sound and complete.*

*Proof.* We split the proof for soundness into 3 parts:

1. There is a satisfying example to $P_{ee}$ if and only if there are exit $k \in ee \cup \{last\}$ and runner-up $i \in \mathcal{C} \setminus \{w\}$ such that the runner-up wins in the exit of the $k$'th layer and there is no preceding layer where the winner wins. In the other direction, there is no counterexample if and only if the negation of the above is true: for every input, either the runner-up does not win or the winner has already won in one of the preceding exits. The negation is encoded in line 5 (for early exit) and in line 7 (for the last exit) in Alg. 1.

2. If Alg. 1 returns UNSAFE result, one of the properties in lines 5 or 7 is UNSAFE. It means that Alg. 1 UNSAFE is sound if the underlying verification tool is sound.

3. Otherwise, if Alg. 1 does not return UNSAFE, it returns SAFE at the last line. Since the algorithm iterates over all exits and in each exit goes through all runner-ups, we can derive from the completeness of the underlying verification tool that if there is no counterexample then $P_{ee}$ is not satisfiable. Therefore, Alg. 1 SAFE answer is sound if the underlying verification tool is complete.

We can conclude that if the underlying verification tool is sound and complete, Alg. 1 is sound and complete too: if UNSAFE is returned, the result is sound, and if SAFE is returned, it is also sound.

Regarding completeness, from the completeness of the underlying verification tool, every query in lines 5 or 7 is guaranteed to be finished in finite time, and there is a finite number of queries, so the whole algorithm is guaranteed to always return either SAFE or UNSAFE in finite time, therefore it is complete. $\square$

### C.2  PROOF OF THEOREM 2

We again define the complexity class *FPT* and give an example.

**Definition C.1** ( (Downey, 2012, Def. 1)). A problem is *fixed-parameter tractable* (*FPT*) with respect to a parameter $p$ if it can be solved in time $f(p) \cdot poly(n)$, where $f$ is a computable function of $p$, and $n$ is the input size.

For example, solving local robustness in a neural network with ReLU activations only is $FPT(k \cdot d)$, where $k$ is an upper bound on the number of neurons in every layer, and $d$ is the number of layers in the network. This is due to the fact that each ReLU activation can be assessed as the active or inactive

case, resulting in $2^{k \cdot d}$ options to define the linear constraints of the ReLUs, and each combination can be solved with linear programming methods in $poly$(#neurons in $\mathcal{N}$). Therefore if we fix the parameters $k, d$ the problem is $poly$(#neurons in $\mathcal{N}$).

**Theorem.** *Given a network $\mathcal{N}$ with EEs and ReLU activations, layer width bound $k$, input $x$, and $\epsilon > 0$, if $\tau(x)$ is stable in $B_\epsilon^x$, then solving $P_{ee}$ with $(\mathcal{N}, x, \epsilon)$ is $FPT(k \cdot |\tau(x)|)$, where $|\tau(x)|$ is the number of layers in $\tau(x)$.*

*Proof.* Given that $\tau(x)$ is stable in $B_\epsilon^x$, the output of every input in $B_\epsilon^x$ is obtained at the same exit as the exit of $\mathcal{N}(x)$. In that case, the verification process can avoid checking all the following layers, and instead check only the layers until the exit where $x$ was obtained. The number of neurons until this exit is limited by $k \cdot |\tau(x)|$. Solving the query can be done by splitting each ReLU into two cases - active ($y = x$) and inactive ($y = 0$) - and the complexity of solving a set of linear constraints which is polynomial in the number of neurons in $\mathcal{N}$ is $poly$(#neurons in $\mathcal{N}$). The number of choices for the activations of the neurons is $2^{k \cdot |\tau(x)|}$. Therefore, the problem is in $FPT(k \cdot |\tau(x)|)$. $\qquad\square$

### C.3 PROOF OF THEOREM 3

We separate the two independent optimizations *break* and *continue* applied in Alg. 2 one after the other into two algorithms: Alg. 4 (which include only the orange lines in Alg. 2) and Alg. 3 (which include only the blue lines in Alg. 2). We prove soundness and completeness for each of the algorithms (by proving the equivalence of each of them to Alg. 1), and then derive the correctness of Thm. 3 from both.

---

**Algorithm 3** Verify DNNs with Early Exits - *Break* Optimization

---

**Input** $\mathcal{N}, x, \epsilon_p$ **Output** $\mathcal{N}$ is robust in $B_\epsilon(x)$, or counterexample

1: $w = argmax(\mathcal{N}(x))$
2: $ee :=$ indices of layers with early exits in $\mathcal{N}$
3: $last :=$ index of last layer in $\mathcal{N}$
4: **for** $k \in ee \cup \{last\}$ **do**
5:     **if** k $\neq$ last **then**
6:         $\mathcal{P} := \exists x' \in B_\epsilon^x : \mathcal{N}(x')_w^k < T$
7:     **else**
8:         $\mathcal{P} := \exists x' \in B_\epsilon^x, \exists i \in \mathcal{C} \setminus \{w\} : \mathcal{N}(x')_w < \mathcal{N}(x')_i$
9:     **end if**
10:   res, cex = Verify($\mathcal{N}, B_\epsilon^x, \mathcal{P}$)
11:   **if** res == SAFE **then**
12:       **return** SAFE
13:   **end if**
14:   res, cex = $ExistsPrevCEX(\mathcal{N}, x, \epsilon, k)$
15:   **if** res == UNSAFE **then**
16:       **return** UNSAFE, cex
17:   **end if**
18: **end for**
19: **return** SAFE

---

**Theorem 6.** *Alg. 4 is equivalent to Alg. 1.*

*Proof.* The additional logic in Alg. 4, highlighted in orange, is introduced in line 5. It checks whether the winner's value might be smaller than $1 - T$. If this condition is not satisfied, no runner-up can exceed $T$, given that the sum of all class values equals 1. Consequently, no counterexample exists, so the result of $ExistsPrevCEX(\mathcal{N}, x, \epsilon, k)$ must be SAFE, and we can skip it and continue to the next iteration of the *for* loop in line 4, which correspond to skipping on one iteration of the *for* loop in lines 3-13 in Alg. 1).

If the condition is satisfied, or in the case of the last layer (where the winner is the maximum value, providing no assurance that a runner-up does not win even if the winner always exceeds $1 - T$), the loop cannot be skipped. In these scenarios, the verification process proceeds equivalently to the processes in Alg. 1 and Alg. 3.

---

**Algorithm 4** Verify DNNs with Early Exits - *Continue* Optimization

---

**Input** $\mathcal{N}$, $x$, $\epsilon_p$ **Output** $\mathcal{N}$ is robust in $B_\epsilon(x)$, or counterexample

1: $w = argmax(\mathcal{N}(x))$
2: $ee$ := indices of layers with early exits in $\mathcal{N}$
3: last := index of last layer in $\mathcal{N}$
4: **for** $k \in ee \cup \{last\}$ **do**
5:     res, cex = Verify($\mathcal{N}, B_\epsilon^x, \exists x' \in B_\epsilon^x : \mathcal{N}_w^k(x') < 1 - T$)
6:     **if** k == last $\vee$ res == UNSAFE **then**
7:        res, cex = $ExistsPrevCEX(\mathcal{N}, x, \epsilon, k)$
8:        **if** res == UNSAFE **then**
9:           **return** UNSAFE, cex
10:        **end if**
11:     **end if**
12: **end for**
13: **return** SAFE

---

Overall, in both cases where the condition is met or not, the result is equal to the result obtained by Alg. 1, as required. □

**Theorem 7.** *Alg. 3 is equivalent to Alg. 1.*

*Proof.* Alg. 3 introduces an additional condition in each iteration to check whether the original winner might not win in the current exit. If this condition cannot be satisfied (SAFE is returned and the condition in line 11 holds), the verification process halts, and SAFE is returned. This is valid because, for all subsequent exits, neither line 5 nor line 7 of Alg. 1 would hold, as there exists a previous exit (the current one) where the winner wins for every example. Consequently, Alg. 1 would also return SAFE in this scenario.

If the condition is satisfiable, the condition in line 11 does not hold and Alg. 3 continues to line 14 to check if there is a counterexample where a runner-ups wins, just as Alg. 1 does in the *for* loop in lines 3-13. If such an example is found and UNSAFE is returned, both algorithms return UNSAFE (line 16 in Alg. 3 and line 13 in Alg. 1). If no counterexample was found for any runner-up in any exit, SAFE is returned in both algorithms (last line), ensuring they are equivalent in their results. □

We can now prove the correctness of Theorem 3.

**Theorem.** *Alg. 2 is equivalent to Alg. 1.*

*Proof.* Alg. 2 sequentially incorporates the optimizations in both Alg. 4 and Alg. 3. Consequently, and based on Thm. 6 and Thm. 7, Alg. 2 is equivalent to Alg. 1.

□

# D   DATASETS AND MODELS TECHNICAL DETAILS

Below, we provide a brief description of each dataset and architecture, summarize their key characteristics in Tab. 4 and Tab. 5, and present the full training protocol used in our experiments.

We used three common datasets:

- **MNIST** (LeCun et al., 2010): A dataset of handwritten digits consisting of grayscale images. This dataset is widely used for evaluating classification methods due to its simplicity and accessibility, featuring 10 classes (digits 0-9).
- **CIFAR-10** (Krizhevsky, 2009): A dataset comprising color images, categorized into classes such as airplanes, cats, and trucks. It is a standard benchmark for formal verification of image classification tasks.
- **CIFAR-100** (Krizhevsky, 2009): A more challenging extension of CIFAR-10, featuring more classes with fewer samples per class, increasing the complexity of the classification.

Table 4: Metadata for datasets used in the evaluation.

| Dataset | Train Size | Test Size | #Classes | Input Shape |
|---|---|---|---|---|
| **MNIST** | 60,000 | 10,000 | 10 | $1 \times 28 \times 28$ |
| **CIFAR-10** | 50,000 | 10,000 | 10 | $3 \times 32 \times 32$ |
| **CIFAR-100** | 50,000 | 10,000 | 100 | $3 \times 32 \times 32$ |

For each dataset, we trained models that incorporate EEs to enable intermediate predictions and verification:

- **Fully Connected (FC-6)**: A fully connected architecture with 6 layers, where the first three layers are equipped with EEs. This model was trained on the MNIST dataset.

- **LeNet-5 (CNN)** (LeCun et al., 1998): A well known architectrue that contains 2 convolutional layers forllowed by 3 linear layers. We trained it on CIFAR-10 and added EEs after the first and second convolutional layers.

- **Modified ResNet-18** (He et al., 2016): A ResNet-18 architecture, adapted by replacing MaxPool operations with AveragePool operations. This model was trained on the CIFAR-10 dataset, and early exist where added after the first and second blocks.

- **VGG-16** (Simonyan & Zisserman, 2015): A standard VGG-16 architecture of 13 convolutional layers, partially separated with Adaptive Average pool layers (and not Maxpool layers, for the reason explained in the last clause), and followed by 3 linear layers. We trained it on CIFAR-10, and incorporated EEs after the 6th and 10th convolutional layers.

Table 5: Characteristics and Evaluation Metrics of Different Models.

| Model | Dataset | Size | # Layers | Accuracy | EE Accuracy | Exit Distribution |
|---|---|---|---|---|---|---|
| **FC-6** | MNIST | 1,519,720 | 6 | 98.18 | 98.2 | [9772, 152, 46, 30] |
| **CNN** | CIFAR-10 | 596,178 | 5 | 70.02 | 69.93 | [5924, 2066, 2010] |
| **ResNet-18** | CIFAR-10 | 11,243,102 | 18 | 86.43 | 86.11 | [6656, 1899, 1445] |
| **VGG-16** | CIFAR-10 | 33,769,566 | 13 | 93.45 | 93.14 | [8268, 1278, 454] |

**Training Protocol**   All models are trained using standard supervised learning on their respective datasets. For CIFAR-10 and CIFAR-100, we apply data augmentation (random cropping and horizontal flipping) and normalize using dataset-specific statistics. **MNIST** models are trained for 10 epochs using SGD, with the learning rate reduced by a factor of 10 every 4 epochs. **CIFAR-10 models** (ResNet-18, VGG, and LeNet variants) are trained for 200 epochs (30 for LeNet). ResNet and VGG use SGD with momentum 0.9, weight decay $5 \times 10^{-4}$, and a step-based learning rate schedule that reduces the learning rate by a factor of 0.1 at epochs 100 and 150. LeNet uses the Adam optimizer with an initial learning rate of 0.001, decayed by 0.1 at epochs 10 and 20. **CIFAR-100 models** use the Adam optimizer with an initial learning rate of 0.001 and cosine annealing over 200 epochs. For the early-exit heads, we add an extra fine-tuning phase: we freeze all backbone parameters and train each exit sequentially for a small number of epochs (20 for larger models, 10 for FC and LeNet), using the same optimizer and initial learning rate as the base model. The learning rate is decayed twice by a factor of 0.1 during this phase.

**Technical Contribution and Non-Triviality.**   While our method is conceptually simple, we view this as an advantage rather than a limitation. Importantly, two technical aspects highlight its non-triviality. First, our approach introduces many additional verification queries in the worst case, and its effectiveness depends on whether early exits succeed in verification. Networks augmented with early exits often achieve faster inference but may sacrifice robustness, which directly impacts verification performance. Thus, our method entails a risk/value tradeoff - performance gains are not guaranteed and are validated only empirically. Second, each partial verification query is not merely a smaller subproblem of the original one; it verifies a more complex property involving the robustness of the exit condition, including Softmax. This distinction makes the problem strictly harder, and the benefit

of our approach is not obvious a priori. In fact, it fails in tools that lack proper Softmax support. The novelty of our contribution lies in showing that, when combined with a state-of-the-art verifier that supports SoftmaxXu et al. (2021); Wang et al. (2021), this strategy does indeed yield improved performance.

**Relation to Certified Training**   Unlike certified training methods, which modifies the training process to improve robustness, our method preserves standard training and instead augments the model with early exits to accelerate verification. This design focuses on reducing verification time rather than altering the robustness–accuracy trade-off. These differences make the two approaches orthogonal: certified training and early exits can be applied independently or even combined within the same network to obtain complementary benefits.

Nevertheless, we provide a comparison to illustrate the performance of our method relative to certified training. Results reported in (Mao et al.) show that CNN-7 with $\epsilon = 2/255 \approx 0.0078$ achieves accuracy 78.82% and robustness 64.41%. In contrast, our early-exit CNN-5 with $\epsilon = 0.005$ and threshold $T = 0.9$ attains accuracy 71.34% and robustness 60.61%, despite relying on a significantly smaller network. These results demonstrate that our fine-tuning preserves robustness while still providing efficiency gains.

## E   Verification Exit Versus Inference Exit: Results for MNIST & FC-6

The details of FC-6 and MNIST are added in Fig. 8. Here, too, there is a high correlation between the verification exit and the inference exit when the result is SAFE, and most of the queries are resolved in the first exit when the result is UNSAFE.

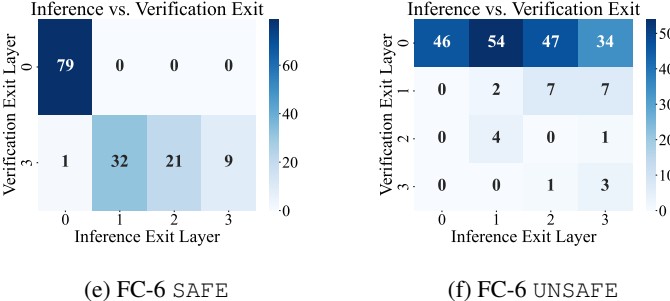

(e) FC-6 SAFE          (f) FC-6 UNSAFE

Figure 8: Heatmaps demonstrating the correlation between the inference and verification exit layers for FC-6 and MNIST.

## F   Comparing the Algorithms and the Optimizations

In this section we compare Alg. 1 and Alg. 2 with all or part of the optimizations (in blue and orange lines) as an ablation study. Following App. C, we denote the variant with the *break* optimization only (the blue lines) with Alg. 3, and the variant with the *continue* optimization only (the orange lines) with Alg. 4. Fig. 9 compares the evaluation results of algorithm pairs across benchmarks, where the x-axis and y-axis represent the runtime (in seconds) of the respective algorithms. Points are color-coded based on the experiment results.

Subfigures (a)-(c) in Fig. 9 compare algorithms 1 through 4 on MNIST, showing that (a) Alg. 3 outperforms Alg. 1, (b) Alg. 4 outperforms Alg. 3, and (c) Alg. 2 outperforms Alg. 4. While (b) highlights that no single heuristic consistently outperforms the other, (c) demonstrates that the combined method always excels in SAFE cases, with negligible additional runtime in simpler cases. After establishing that the combined algorithm is the optimal choice, subfigures (d)-(f) compare Alg. 1 and Alg. 2 on three additional benchmarks: CNN, ResNet-18, and VGG-16, all trained on CIFAR-10. These results highlight the superiority of the improved algorithm in SAFE cases while

maintaining equal runtimes for `UNSAFE` cases. Note that in the VGG-16 case, the original network failed to produce `SAFE` results, whereas the modified network with EEs succeeded. This discrepancy explains why (f) compares only the `UNSAFE` results.

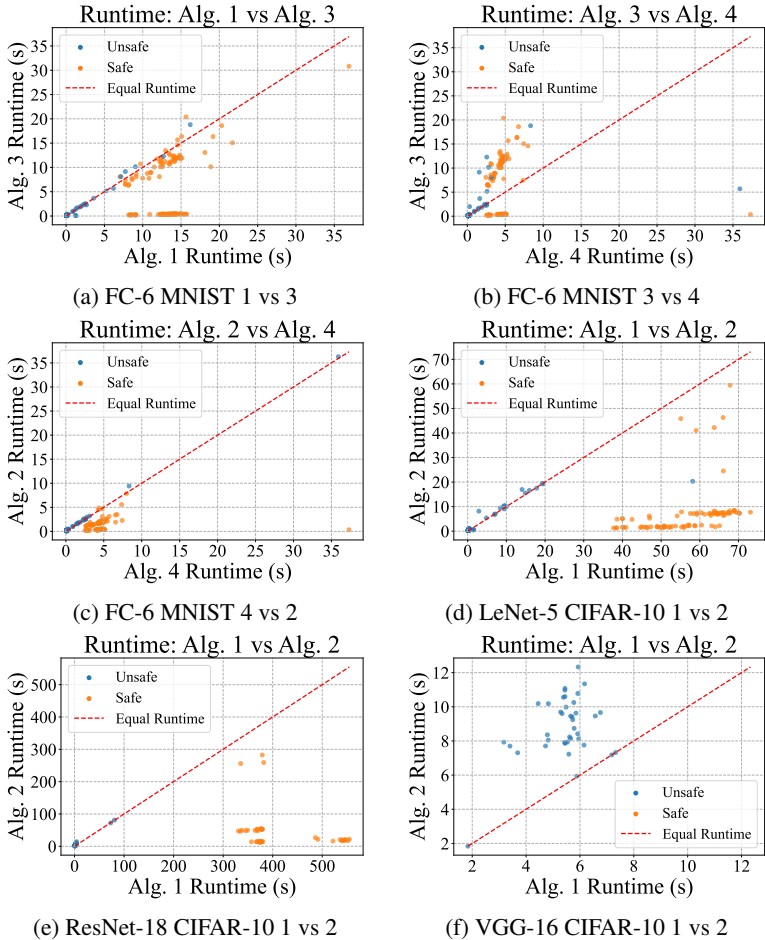

Figure 9: Comparing algorithms 1, 2, 3, 4 over various benchmarks. Graphs (a-c) demonstrates the ablation study on the optimization: the break optimization (Alg. 4) improves the basic algorithm (a), and the continue optimization improves it even further for MNIST with FC-6 (b), but applying both optimization is the best option (c). In graphs (d-f) we compare Alg. 1 and Alg. 2 on other datasets.

## G    EXTENDABILITY TO OTHER PROPERTIES AND ACTIVATION FUNCTIONS

Our framework is not limited to local robustness. While we instantiated it for robustness to keep the presentation focused and experimentally tractable, the same formulation naturally extends to other property types. Formally, at each exit $k$, the robustness clause can be replaced by an arbitrary predicate $\phi$ over the network outputs:

$$\exists x' \in B_\varepsilon(x) : \phi(N_k(x')) \wedge \forall e < k : N_e^w(x') < T_e,$$

where the right-hand term enforces the early-exit semantics (i.e., no earlier exit fires).

Examples include:

1. Safety constraints: $\phi \equiv g(N_k(x')) \leq 0$, representing, for instance, bounded control actions or adherence to physical or operational limits.

2. Fairness predicates: $\phi \equiv |N_k(x'_a) - N_k(x'_b)|_\infty \leq \delta$ or $\arg\max N_k(x'_a) = \arg\max N_k(x'_b)$, where $x'_a$ and $x'_b$ are counterfactual inputs differ only in sensitive attributes (e.g., gender or race).

The Break heuristic generalizes naturally by substituting the class-margin test with corresponding sufficient conditions for the new property. Since only the inner predicate $\phi$ changes, the overall control flow, soundness, and fixed-parameter-tractability guarantees remain valid for any activation function. The only exception is the Continue optimization, which was specifically tailored to the robustness setting and may need adjustment or removal for other properties. Fig. 9 at App. F shows the effectiveness of the framework when only the Break heuristic is applied.

Our framework is activation independent. The theoretical result in Thm. 2 assumes piecewise-linear activations to ensure that each verification query can be split into a finite number of polynomial-time subproblems, but this assumption is not essential to the soundness and completeness of the general formulation, as long as the activation functions are supported by the underlying verification tool.

Lastly, our soundness and completeness guarantees for both the basic solution and the Break optimization are unaffected by altering either the verified property or the network's activation functions. We deliberately focused on local robustness to provide a clear theoretical foundation and a practical benchmark. Extending the framework to safety or fairness specifications mainly requires integrating the corresponding domain-specific encodings and datasets, while only minor adaptations of the Continue optimization are expected - future directions that are natural rather than conceptual challenges. While other property types are not our current focus, the feedback you raised would help us formulate a generalized version of the problem.

## H    HARDWARE VALIDATION

We clarify our hardware setup and provide an additional experiment on a conventional GPU platform. All main experiments were executed on a recent **Apple M3** machine with an integrated **8-core GPU** (specifications noted in 4.1). This environment was shared across all baselines, and the $\alpha\beta$-CROWN verifier was run faithfully. To verify that our conclusions are not tied to the M3 architecture, we repeated the CIFAR-10 ResNet-18 experiment from Fig. 2(b) on an **NVIDIA A100**. Table 6 reports the results and shows that the same improvement trend holds across both hardware platforms.

Table 6: Total verification time (in hours) on Apple M3 and NVIDIA A100 for CIFAR-10 ResNet-18, with and without EEs, split by SAFE and UNSAFE queries.

| Condition | M3 (h) | A100 (h) | Acceleration |
|---|---|---|---|
| No-EE SAFE | 1.5533 | 1.3144 | 15.38% |
| EE SAFE | 0.3721 | 0.2839 | 24.00% |
| No-EE UNSAFE | 0.0378 | 0.0261 | 30.93% |
| EE UNSAFE | 0.0579 | 0.0410 | 28.68% |

The improvement trend is identical across hardware. SAFE queries: EEs reduce runtime by $4.17\times$ on M3 and $4.63\times$ on A100. UNSAFE queries: ratios are similar, and runtimes are short. EE speedup is even larger on A100 (24% vs. 15.4%). These results confirm that our method's benefits are **hardware-independent**.

## I    DISCLOSURE: USE OF LARGE LANGUAGE MODELS (LLMS)

The authors were solely responsible for developing the research questions, designing the methodology, performing the analysis, and interpreting the findings. A large language model (LLM) was employed only to assist with improving the clarity and style of the writing, without influencing any substantive aspects of the research.

