# OpenReview forum: "Bridging Efficiency and Safety: Formal Verification of Neural Networks with Early Exits"
_ICLR.cc/2026/Conference — Submitted to ICLR 2026_

### Official Review · Reviewer_KfRp · 2025-10-26

**Soundness:** 3
**Presentation:** 3
**Contribution:** 3
**Rating:** 6
**Confidence:** 4

**Summary:**

The paper formalizes local robustness for networks with early exits (EEs) by redefining the robustness counterexample to account for conditional execution and multiple potential output layers. It proposes: (i) a baseline verification procedure (Alg. 1) that mirrors EE inference logic and is sound and complete given a sound/complete backend, and (ii) an optimized procedure (Alg. 2) with “break/continue” checks that can certify safety earlier and skip many inner-loop queries. The authors also give a trace-stability notion and show fixed-parameter tractability under this assumption (FPT in layer width × length of the stable trace). Experiments on MNIST/CIFAR (FC, LeNet, ResNet-18, VGG-16) show large runtime gains for SAFE cases (up to ~10×) and that adding EEs to standard models can make them easier to verify without large accuracy loss; they also analyze threshold/exit-placement trade-offs.

**Strengths:**

1. Precise redefinition of robustness with EEs and proofs of soundness/completeness for both algorithms; FPT analysis under trace stability gives theoretical insight.


2. Alg. 2’s early break and continue checks skip redundant exit evaluations once safety or unsafety is determined, greatly speeding up SAFE certifications (e.g., VGG-16 cases that previously timed out).

3.  Adding EEs often accelerates verification (particularly SAFE instances) and exit placement/thresholds provide a tunable accuracy-latency-verifiability trade-off.

**Weaknesses:**

1. Because α/β-CROWN lacked nested AND encoding, queries were decomposed into multiple conjunctive calls; this may distort absolute runtimes and comparability to “vanilla” queries. A native-AND backend (e.g., Marabou) could change results.


2. Scope restricted to local robustness and ReLU. Theory and claims lean on ReLU and local robustness; no formal handling of other properties (e.g., fairness/safety specs) or activations beyond brief discussion.



3. Runs are on a single M3 machine with modest samples; many UNKNOWN arise from timeouts/loose bounds. More large-scale, multi-backend evidence (and variance/CI reporting) would strengthen the case.


4. The “break/continue” conditions are intuitive but largely justified empirically; a tighter analysis of when these checks dominate (beyond the trace-stability assumption) would help.

**Questions:**

1. A recent paper, “Learning Minimal Neural Specifications” (Geng et al., NeuS 2025), investigates neural activation patterns (NAPs) and shows that only a small subset of neurons is often sufficient to characterize a model’s robust behavior. Your early-exit mechanism seems conceptually related—it also leverages intermediate representations that can certify outputs earlier. Do you see these two phenomena as reflecting the same underlying principle of redundant or concentrated information flow in deep networks? Could you discuss the connections to Geng et al.?

2. Beyond local robustness: How would you instantiate your framework to other properties (e.g., safety constraints or fairness predicates) within the EE setting? What changes are needed in the property encoding and in your early checks?

---

> ### Author Response · Authors · 2025-11-17
>
> We thank the reviewer for their thoughtful feedback, especially for recognizing our formal redefinition of robustness with EEs, the accompanying proofs, the FPT analysis under trace stability, and the practical gains of Algorithm 2’s early checks together with the tunable accuracy-latency-verifiability trade-off enabled by exit placement and thresholds.
>
> In the following, we address all weaknesses (W) and questions (Q) raised.
>
> **#W1:**
>
> We also evaluated Marabou and got some SAFE (“UNSAT” in Marabou) results for queries with small $\epsilon$ values, but it failed to finish solving even the query on the first early exit (i.e. the verification process was stuck). After consulting the maintainers, we learned that, although Marabou can encode SoftMax constraints, its practical handling of the exponential-normalization pattern remains fragile and limits its effectiveness on these queries. We believe that for small epsilons, the bound propagation phase of Marabou works even on SoftMax, but for a bit harder queries where full verification applies, it stucks. This is the only reason why we present results only with ABCrown. Other verifiers, such as NeuralSAT (VNNCOMP`24 runner-up) lack support for SoftMax [A, Section 5], so we could not use it standalone under our encodings.
>
> **#W2, #Q2:**
>
> We have included a generalized formulation in Appendix G to handle this important point.
>
> (1) *Our framework is not limited to local robustness.* While we instantiated it for the most explored property of local robustness to keep the presentation focused and experimentally tractable, the same formulation naturally extends to other property types. Formally, at each exit $k$, the robustness clause can be replaced by an arbitrary predicate $\phi$ over the network outputs:
> $\exists x'\in B_\varepsilon(x): \phi(N_k(x')) \land \forall e<k: N_e^w(x')<T_e$,
> where the right-hand term enforces the early-exit semantics (i.e., no earlier exit fires).
>
> Examples include:
> - Safety constraints: $\phi \equiv g(N_k(x')) \le 0$, representing, for instance, bounded control actions or adherence to physical or operational limits.
> - Fairness predicates: $\phi \equiv \lVert N_k(x'_a) - N_k(x'_b)\rVert _\infty \le \delta$ or $argmax(N_k(x'_a)) = argmax(N_k(x'_b))$, where $x'_a$ and $x'_b$ are counterfactual inputs differing only in sensitive attributes (e.g., gender or race).
>
> The Break heuristic generalizes naturally by substituting the class-margin test with corresponding sufficient conditions for the new property. Since only the inner predicate $\phi$ changes, the overall control flow, soundness, and fixed-parameter-tractability guarantees remain valid for any activation function. The only exception is the Continue optimization, which was specifically tailored to the robustness setting and may need adjustment or removal for other properties. We already show in Appendix F (Fig. 9) that the framework remains effective even when only the Break heuristic is applied.
>
> (2) *Our framework is activation independent.* The theoretical result in Thm. 2 assumes piecewise-linear activations to ensure that each verification query can be split into a finite number of polynomial-time subproblems, but this assumption is not essential to the soundness and completeness of the general formulation, as long as the activation functions are supported by the underlying verification tool.
> We deliberately focused on local robustness, which is the most common setting for studying fundamental DNN verification algorithms [B,C,D,E]. This focus provides a clear theoretical foundation and a practical benchmark for evaluating algorithmic soundness and scalability. Extending the framework to safety or fairness specifications would mainly require integrating the corresponding domain-specific encodings and datasets, while only minor adaptations of the Continue optimization are expected - future directions that are natural rather than conceptual challenges. For instance, safety verification approaches [F,G] and fairness verification methods [H,I] can be instantiated in our framework by substituting their respective constraint formulations for the robustness clause at each exit. While other property types are not our current focus, the feedback you raised would help us formulate a generalized version of the problem.
>
> **#W3:**
>
> Please refer to our response to Reviewer upVP (#W5, #Q2), where we provide a detailed answer and include a direct comparison on an NVIDIA A100 that reconfirms our results.

---

> > ### Author Response · Authors · 2025-11-17
> >
> > **#W4:**
> >
> > We thank the reviewer for this insightful point. The theoretical and empirical justification for the Break/Continue conditions is already included in the appendix. In Appendix C, we formally separate and prove the soundness and completeness of the two optimizations (Algorithms 3 and 4). Appendix F then provides an ablation study showing when each dominates: the Break rule eliminates redundant exit checks once a property is determined SAFE, while the Continue rule skips subsequent verifications under stable traces. Together, they yield substantial runtime gains (up to ×2–3 speedups on SAFE cases in MNIST/CIFAR), confirming that these heuristics are not ad-hoc but follow directly from the structure of trace-stable executions.
> >
> > **#Q1:**
> >
> > Thank you for pointing us to this recent and closely related work by Geng et al.; we now cite and discuss it in the revised manuscript. Their study and ours are driven by a similar underlying insight: deep networks often concentrate decisive information within a small, stable portion of their computation. Geng et al. reveal this through minimal neural activation patterns (NAPs), while we expose it through early exits, where intermediate layers already contain sufficient evidence for both prediction and verification.
> > Conceptually, both approaches share a common structure: for inputs satisfying some stability pattern, a stronger property follows. In our case, trace-stable inputs consistently take the same exit, allowing verification to succeed on the corresponding prefix. In their case, inputs belonging to the same semantic/robust class share a minimal NAP.
> > The key distinction lies in the nature of the guarantees and the objective. Geng et al. focus on discovering statistical specifications from activation patterns, whereas our method provides sound and complete formal verification along conditional execution paths. Their approach identifies minimal neuron subsets across the entire network; ours identifies safe stopping points in the computation and formally proves correctness at each exit.
> > Importantly, the two perspectives are complementary. NAPs could inform where early exits are most effective or which neurons within an exit contribute most to its reliability, while our trace-based verification framework offers a principled way to certify the robustness of such minimal subsets.
> > We highlight this relationship and its potential synergy in the revised Related Work section (lines 493–499), where we also include the relevant citations [J, K].
> >
> > A. A DPLL(T) Framework for Verifying Deep Neural Networks, arXiv 2024.
> >
> > B. Scalable Neural Network Verification with Branch-and-bound Inferred Cutting Planes, NeurIPS 2024.
> >
> > C. Tightening Robustness Verification of MaxPool-based Neural Networks via Minimizing the Over-Approximation Zone, CVPR 2025
> >
> > D. Beta-CROWN: Efficient Bound Propagation with Per-Neuron Split Constraints for Neural Network Robustness Verification, Wang et al., NeurIPS 2021.
> >
> > E. Algorithms for Verifying Deep Neural Networks, Liu et al., FnT Optimization 2021.
> >
> > F. AI2: Safety and Robustness Certification of Neural Networks with Abstract Interpretation, Gehr et al., IEEE SP 2018.
> >
> > G. Verification of LSTM Neural Networks with Non-linear Activation Functions, Moradkhani et al., NFM 2023.
> >
> > H. Fairness Through Awareness, Dwork et al., ITCS 2012.
> >
> > I. Fairness as a Program Property, Albarghouthi et al., CAV 2017.
> >
> > J. Towards Reliable Neural Specifications, ICML 2023.
> >
> > K. Learning Minimal Neural Specifications, NeuS 2025.

---

> ### Comment · Reviewer_KfRp · 2025-11-18
>
> I appreciate the authors’ detailed rebuttals and clarifications to my questions. The work remains interesting and is supported by both theoretical analysis and empirical evidence. I will therefore maintain my positive evaluation and score.

---

> > ### Author Response · Authors · 2025-11-20
> >
> > Thank you very much for taking the time to engage deeply with our responses. We appreciate your positive evaluation and are glad that our clarifications addressed your concerns. Your feedback helped us improve the paper, and we are grateful for your support.

---

### Official Review · Reviewer_6kQJ · 2025-10-29

**Soundness:** 3
**Presentation:** 3
**Contribution:** 2
**Rating:** 4
**Confidence:** 4

**Summary:**

The paper explores how formal verification techniques can be extended to neural networks with early exits. The authors propose a robustness definition that accounts for conditional execution paths and introduce two verification algorithms: a baseline and an optimized version with early-stopping heuristics. They provide proofs of soundness and completeness and evaluate their approach on MNIST, CIFAR-10, and CIFAR-100 using various architectures.

**Strengths:**

* Tackles an underexplored intersection between neural networks with early exits and formal verification.
* Theoretical development, proofs, and algorithms are clear and internally consistent.
* Experiments are well executed across multiple datasets and architectures, supporting the main claims.
* The presentation is clean and well structured, making it easy to follow the core ideas.

**Weaknesses:**

* The novelty is somewhat limited. The work mainly adapts existing verification methods to networks with conditional exits, which, while useful, feels like an incremental extension rather than a conceptual breakthrough.
* While the initial idea of applying formal verification to early-exit networks is well motivated, many of the subsequent algorithmic and analytical components follow naturally from that setup and feel more like straightforward extensions than new conceptual contributions.
* A large fraction of UNKNOWN results in Figure 1 indicates that the method still faces scalability and solver limitations.

**Questions:**

The experiments analyze the effect of the confidence threshold $T$ empirically, but could you comment on how sensitive the formal guarantees are to small variations in $T$? For instance, would a slight change in the threshold invalidate previously proven robustness properties?

---

> ### Author Response · Authors · 2025-11-17
>
> We thank the reviewer for recognizing the novelty of addressing the intersection between early-exit networks and formal verification, the clarity and consistency of our theoretical development and proofs, and the strength and clarity of our experimental evaluation and presentation.
>
> In the following, we address all weaknesses (W) and questions (Q) raised.
>
> **#W1**
>
> We agree that the high-level idea - verifying networks with early exits - is natural once EEs exist. We also agree that the technical steps needed to define $P_{ee}$ in a form compatible with standard verifiers are not overly complicated, though they are still not trivial. However, we want to emphasize that this extension is conceptually non-obvious, because **verification queries in EE networks require redefining robustness itself**. There are two *conceptual* gaps that do not appear in ordinary networks:
>
> 1. **Conditional prediction constraints.** Early-exit inference involves a conditional control flow, and it is not clear a priori how a single verification query can capture this process. The key insight behind our formulation is that each exit can be verified through an **equal condition**: “the winner must remain the winner until exit $i$ and must trigger exit $i$.” This observation allows us to define a clean and uniform property without combining many unrelated clauses.
>
> 2. **Multiple competing outputs at different depths.** Each exit exposes its own decision point, and ruling out all alternative classes at multiple depths is fundamentally different from verification in a single-head network.
> Without redefining the robustness property $P_{ee}$ and its encoding, robustness for EE networks is not even well-posed. Our formulation addresses these gaps and ensures classical verifiers can be applied at all.
>
> **#W2:**
>
> We thank the reviewer for highlighting this important point. We are glad that our presentation makes things clear and readable. However, there are multiple subtle points that our solution and optimization address. Below is an example for two similar cases one has to consider for Alg. 1 design, each requires a careful different handling. We hope this example partially demonstrates the technical depth of our solution:
>
> 1. Spurious SAFE.
> Obtaining a SAFE result for the constraint $N(x')^e_w > T_e$ at exit $e$ does not necessarily mean that we can apply the Break optimization and stop verification. We must ensure that no earlier exit invalidates the property. Our contribution is the insight that it is sufficient to check only the previous exit $(e-1)$ to soundly conclude that the SAFE result at exit $e$ is valid.
>
> 2. Spurious UNSAFE.
> Similarly, an UNSAFE result must be handled with great care. The returned counterexample is not necessarily valid for the model, because it may violate an earlier exit condition and therefore never reach exit $e$ during inference. Unlike the SAFE case, here we must re-check the counterexample against **all** earlier exits to ensure correctness.
>
> In addition, our method does not simply apply “straightforward subsequent algorithms.” The Continue optimization (Algorithm 2, lines 12–13) is tailored specifically to the local robustness property and is not an immediate or generic improvement. It relies on the key observation $N(x’)^k_w>1-T \implies \forall r\in\mathcal{C}\setminus\{w\}N(x’)^k_r<T$, which allows us to soundly skip many verification queries. Figure 9 in Appendix F demonstrates the independent contribution of this optimization.
>
> The analytical component of the work is also non-trivial. We explain why our approach is efficient not only for easy queries but also for hard ones. We show that (1) **theoretically**, under the *trace-stability* assumption, the complexity of our method improves, and (2) **empirically**, this assumption holds in practice. Together, these results explain the efficiency of our approach.
>
> Finally, we show for the first time that early exits - introduced for inference efficiency - can also be leveraged to accelerate formal verification. Our experiments demonstrate **substantial runtime improvements compared to a state-of-the-art verifier**, and our theoretical analysis explains these gains via the trace-stability hypothesis.

---

> ### Author Response · Authors · 2025-11-17
> **Official Comment by Authors**
>
> **W3 - On UNKNOWN cases.**
>
> This is a long-standing challenging problem, and previous works typically have many UNKNOWN instances [37,38]. While formal verification of DNNs is an NP-complete problem, the scalability of practical verifiers has steadily improved in recent years. Our results demonstrate substantial gains over AB-CROWN, a state-of-the-art verification tool, including in many instances where scalability is a known bottleneck (see lines 1126-1137 for a detailed discussion). Moreover, because our method verifies properties at early layers, and networks with early exits are specifically trained to optimize the accuracy–inference-time tradeoff, many inputs naturally exit in the first layers or blocks. In these cases, our approach is highly scalable, especially when compared to verifying the full network without early exits. We also refer the reviewer to our answer for reviewer upVP’s #W6.
>
> **Q1 - Sensitivity of the formal guarantees to small changes in $T$.**
>
> The robustness guarantees produced by our framework are theoretically stable under small variations of $T$. This follows directly from the soundness and completeness of the reduction: the property we verify is monotone in $T$. Concretely:
>
> 1. If the verifier returns UNSAFE for a given threshold $T$, the same counterexample $x’$ (for which it holds that $N(x’)^e_r>T$ for some exit $e\in ee$ and runner $r\in\mathcal{C}\setminus\{w\}$) also violates the property for any smaller threshold (if $T’<T$ then $N(x’)^e_r>T>T’$). Therefore, a decrease in $T$ cannot turn an unsafe instance into a safe one.
> 2. If the verifier returns SAFE for $T$, then increasing $T$ only tightens the exit condition and cannot create a new adversarial example that did not already exist: For every input sample, exit index, and runner $(x', e, r)$, if $N(x')^e_r < T$, then for any $T' > T$ we still have $N(x')^e_r < T < T'$. In other words, the runner did not violate the margin at $T$, and therefore it cannot violate it at any larger threshold $T'$. Thus, a small increase in $T$ cannot invalidate an existing safe certificate.
>
>
> In other words, under an ideal verifier with exact arithmetic, the verified property is monotone:
>  $$
>  T_1 < T_2 \implies P_{ee}(T_2) \Rightarrow P_{ee}(T_1), \qquad
>  \neg P_{ee}(T_1) \Rightarrow \neg P_{ee}(T_2).
>  $$
> where $P_{ee}(T_i)$ represents that $P_{ee}$ is SAFE with respect to $T_i$.
>  Hence, slight changes in $T$ cannot invalidate previously established formal guarantees.
> Naturally, practical verifiers operate with finite precision. Our guarantees therefore inherit the usual assumptions about the backend solver (as in all verification pipelines). To check whether such numerical effects could break monotonicity, we ran additional experiments using multiple $\epsilon$ values and many closely spaced values of $T$. Specifically, we selected 25 MNIST samples and evaluated all thresholds $T \in [{0.60, 0.61, \ldots, 0.90}]$ and perturbation radii $\epsilon \in [{0.01, 0.02, 0.03, 0.04, 0.05}]$. Across all these runs, we did not observe a single case where a SAFE result became UNSAFE (or vice versa) solely due to a small change in $T$. The SAFE/UNSAFE outcomes remained consistent throughout.
>
> In all the additional experiments we ran - covering multiple $\epsilon$ values and many very small variations of $T$ - we did not observe any violation of the theoretical monotonicity. The SAFE/UNSAFE outcomes always remained stable across nearby thresholds. This is fully consistent with the mathematical structure of the property, and any deviation would rely solely on numerical precision limitations of the backend solver rather than on our formulation.
>
> Importantly, this behavior also reflects the fact that our reduction is provably sound and complete, a property we designed carefully and formally established. Ensuring soundness and completeness was essential here: it guarantees that the theoretical monotonicity of ($P_{ee}$) with respect to $T$ is faithfully preserved by our method.

---

> ### Author Response · Authors · 2025-11-27
> **Follow-Up Note Regarding Our Updated Response**
>
> Dear reviewer 6kQJ,
>
> As the rebuttal period is close to ending, we kindly ask the reviewer to consider our response and let us know if any further clarification is needed.
>
> Over the past week we have run additional experiments and incorporated the results directly into our updated answers, including revisions to the response regarding the sensitivity to the threshold $T$.
>
> The other reviewers have already responded with positive evaluations, and we hope the final review can also be completed in time.

---

### Official Review · Reviewer_upVP · 2025-10-30

**Soundness:** 2
**Presentation:** 3
**Contribution:** 2
**Rating:** 4
**Confidence:** 5

**Summary:**

This paper introduces the framework for formal verification of neural networks with early exits (EE), addressing the gap between efficiency-oriented EE architectures and formal safety guarantees.  The authors formalize a robustness property tailored to early exit networks that accounts for conditional execution paths and multiple output layers.  They present a baseline verification algorithm along with two key optimizations: early stopping within the verification loop and heuristics to reduce redundant subqueries.

The work is evaluated across multiple datasets (MNIST, CIFAR) and architectures (FC, CNN, ResNet, VGG), demonstrating that networks with early exits can be verified more efficiently than standard networks while maintaining soundness and completeness.

**Strengths:**

- The paper formulates robustness property that  handles the conditional execution logic inherent in EE networks.  This addresses the challenge of multiple potential output layers and conditional branching in verification.
- The theoretical analysis establishes that the problem is Fixed Parameter Tractable (FPT) under trace stability assumptions.  This helps provide  insights into when EE verification becomes useful.
- Results show that adding early exits improves verifiability compared to standard networks.

**Weaknesses:**

- The method only improves runtime of solvable cases, i.e., if the underlying verifier cannot verify a property given sufficient time, neither can the EE approach. From a practical standpoint, verification is typically a one-time cost that can run for days, limiting the real-world utility of runtime improvements. That said, EE would be beneficial from competitions where runtime matters.

- The algorithm scales poorly in the worst case, requiring E·(C-1)+1 verification queries (E is #  exits and C # of classes). This can lead to significant overhead when EE conditions are not met, making verification slower than standard approaches.

- Theoretical guarantees rely heavily on the trace stability assumption, which may not hold in practice. The paper assumes this holds but provides limited analysis of fallback strategies. The FPT complexity result (Thm 2) becomes meaningless if trace stability fails.

- Evaluation should include additional verifiers, e.g., the top X from the DNN verification competition (VNN-COMPs). Most of them are opensource.   The paper also say Marabou supports the format naturally, why not evaluating on it?

- Using ABCrown on an Apple M3 is a very strange setup as ABCrown, as well as most top DNN verification tools, rely on GPU.  In other words, in a proper setup of running DNN verification tools on computer with good GPU, this approach might not provide much improvement.  Running it on M3 appears as if the paper intentionally slow down existing work to highlight their improvements.

- The proposed algorithms do not handle UNKNOWN results from the underlying verifier, which appear frequently in the experimental evaluation.

**Questions:**

- Can early exits actually help verify hard instances that underlying verifiers cannot solve even with extended runtime (e.g., days), or does the method only provide speedups for already solvable cases?

- AB-Crown works best on GPUs, so evaluating on an Apple M3 might not fully disclose the contributions.  Could you run it on a computer with GPU to properly demonstrate the improvement made in this work?

- Evaluating on a single verifier doesn't seem adequate.  There's no reason for not trying other verifiers, which support similar input/output formats as ABCrown.

- UNSAFE results are typically found quickly by verifiers (Fig 2b), yet Fig 2a and 2c show significant runtime for UNSAFE cases in some benchmarks. What accounts for this discrepancy?

- How does the method perform when the trace stability assumption fails?

---

> ### Author Response · Authors · 2025-11-17
>
> We thank the reviewer for recognizing our formulation of robustness for conditional execution in EE networks, the accompanying FPT analysis under trace stability, and the empirical evidence showing that early exits improve verifiability compared to standard networks.
>
> In the following, we address all weaknesses (W) and questions (Q) raised.
>
> **#W1, #Q1:**
>
> Theoretically, every case is solvable, given a sound and complete verifier. In practice, worst case complexity depends on the size of the network being analyzed. Our method substantially reduces the effective network size in many instances (the harder ones), as clearly shown in Figure 2. Consequently, it not only improves the worst-case complexity but also accelerates verification in practice, achieving both benefits simultaneously. Improving the runtime of solvable cases is by itself valuable, as the reviewer noted, but our method also successfully verifies queries that the underlying verifier fails to solve within a reasonable time budget on the original network. We explore this type of queries in our experiments on CIFAR-100, which include a comparison under a 2-hour timeout (a practical and sufficiently generous threshold). As detailed in lines 366–372, while the baseline does not solve any query for epsilon=0.001 (the harder queries), our approach significantly improves the number of solved queries to 14/25 (56%). We attribute this result to the fact that our method verifies queries on substantially smaller networks, making the resulting partial networks solvable in practice even in the presence of extensive branch-and-bound exploration.
>
> **#W2:**
>
> E*(C-1)+1 queries only appear in the worst case in theory, but empirically much fewer queries are needed due to our optimizations. Notice that the Continue optimization effectively filters out many queries even in case that the Break optimization is not applied (Figs. 9(b,c), Appendix F). In addition, the total complexity in a worst case scenario isn’t determined by the number of the underlying verification queries, but the sum of complexities of all the verification queries. Having two simple verification queries (requiring less branch-and-bound) is faster than having one hard verification query that requires lots of branch-and-bound. The verification queries for early exits are smaller than the original verification query with EE. Lastly, despite the entailed risk/value tradeoff in our approach, performance gains are validated empirically. We refer the reviewer to the paragraph “Technical Contribution and Non-Triviality” at Appendix D, lines 1126-1137, where we explicitly mentioned this issue.
>
> **#W3:**
>
> In practice, the trace stability assumption almost always holds for SAFE queries, as demonstrated in Fig. 3 and again in Fig. 8 (in Appendix E): Fig. 3(a) shows that for the ResNet-18 model trained on CIFAR-10, almost 95% (37/39) of the inputs whose inference was finished in the first exit were also verified in the first exit. Fig. 3(c) shows that for the LeNet-5 model trained on CIFAR-10, almost 99% (82/83 inputs) of the inputs whose inference was finished in the first exit were also verified in the first exit. Fig. 8(e) shows that for the FC-6 model trained on MNIST, almost 99% (79/80 inputs) of the inputs whose inference was finished in the first exit were also verified in the first exit. Theorem 2 provides a theoretical analysis that explains the empirical behavior observed in these figures, under the hypothesis that the trace stability assumption holds. Thus, our theory and evaluation are consistent with each other. The assumption does not commonly hold in UNSAFE queries, as expected (since the original sample and counterexamples behave differently, which also undermines the trace stability). It is explicitly mentioned in lines 377-378, and we elaborate on this in our response to #Q4, #Q5 below.
>
> **#W4, #Q3:**
>
> We also evaluated Marabou and got some SAFE (“UNSAT” in Marabou) results for queries with small $\epsilon$ values, but it failed to finish solving even the query on the first early exit (i.e. the verification process was stuck). After consulting the maintainers, we learned that, although Marabou can encode SoftMax constraints, its practical handling of the exponential-normalization pattern remains fragile and limits its effectiveness on these queries. We believe that for small epsilons, the bound propagation phase of Marabou works even on SoftMax, but for a bit harder queries where full verification applies, it stucks. Other verifiers, such as NeuralSAT (VNNCOMP`24 runner-up) lack support for SoftMax [A, Section 5], so we could not use it standalone under our encodings.

---

> > ### Author Response · Authors · 2025-11-17
> >
> > **#W5, #Q2:**
> >
> > We thank the reviewer for raising this point. We conducted all experiments on a recently acquired Apple M3 machine in our lab, equipped with an integrated 8-core GPU (explicitly noted in lines 280-281; full specifications available at Apple Support [B]), and this GPU was fully utilized in all runs. While this setup is less conventional than NVIDIA GPUs, it provided a fair, consistent, and shared environment for all baselines. We emphasize that we did not intentionally slow down any competing method: all verifiers were executed faithfully, and throughout the project we consulted the αβ-CROWN maintainers regarding configuration, runtime parameters, and efficiency issues.
> > To further validate that our improvements are not hardware-dependent, we repeated the CIFAR-10 ResNet18 experiment from Fig. 2(b) on an NVIDIA A100 GPU. Before running, we again sent our configuration to the αβ-CROWN maintainers to ensure optimal settings on that hardware.
> > The following table reports the total verification time (in hours) for all samples. The results confirm the same trend: early exits consistently reduce verification time for SAFE cases, even when both methods run faster overall. Moreover, we can see that (i) In SAFE the ratio with M3 is 1.5533/0.3721=4.174, compared to the ratio in A100 which is 1.3144/0.2839=4.6298. So **the results with A100 are even more in our favor**. (ii) In UNSAFE the ratio with M3 is 0.0378/0.0579=0.6528, compared to the ratio in A100 which is 0.0261/0.0410=0.6366, meaning the change here is small and negligible, especially given the fact that the times for solving UNSAFE queries are relatively short
> >
> > Moreover, the proportional speedup on the A100 is higher for early-exit verification (24% vs. 15.4% in the no-EE case), while UNSAFE queries show similar acceleration (30.9% vs.  28.7%). These results reinforce the original findings and demonstrate that the effect is not hardware-dependent.
> >
> > +----------------------+-----------+-----------+----------------------------+
> >
> > |---  Hardware  ---|  --    M3  --   |- A100  -|  diff (acceleration) |
> >
> > |--    No EE SAFE  --|  1.5533 |  1.3144  |            15.38%      |
> >
> > |-----   EE SAFE  ----| 0.3721  |  0.2839 |            24.00%      |
> >
> > | No EE UNSAFE | 0.0378  |  0.0261  |           30.93%       |
> >
> > |---    EE UNSAFE   --| 0.0579  | 0.0410  |           28.68%       |
> >
> > +----------------------+-----------+-----------+----------------------------+
> >
> > **#Q4, #Q5:**
> >
> > In the vast majority of UNSAFE cases the runtime difference is negligible. There are very few cases in Figs. 2(a,c), where EE Time is bigger, but it still takes a short time. Consistently with our analysis, we believe that in these few cases the trace stability assumption does not hold, and as a result, the verification exit is not equal to the inference exit. In such cases, the “break” optimization does not shorten runtime. If the “Continue” optimization successfully filters out other queries, the overhead is smaller. Otherwise, line 14 of Alg. 2 is executed, and more queries are to be solved. Nevertheless, empirically, Figs. 3 (a,c) demonstrate that the assumption holds for most of the SAFE queries, and it is also demonstrated that this is a pretty common case for UNSAFE results. To improve other UNSAFE cases, we can start the verification in the EE case with adversarial attacks for all exits in parallel. This approach will heuristically remove the few cases where UNSAFE is easily found in the baseline but is not found in EE at the start stage of the verification process.
> >
> > **#W6:**
> >
> > Our method preserves the number of UNKNOWN outcomes in experiments where most queries are successfully solved, and it reduces the number of UNKNOWN results compared to the baseline where the proportion of UNKNOWNs is relatively high. UNKNOWN is empirically very common in previous NN verification works [37,38]. Using a sound and complete verifier, an UNKNOWN result simply means that the query was not solved within the time limit and does not indicate any theoretical limitation.
> >
> > A. A DPLL(T) Framework for Verifying Deep Neural Networks, arXiv 2024.
> >
> > B. https://support.apple.com/en-il/118551

---

> > > ### Comment · Reviewer_upVP · 2025-11-19
> > >
> > > Thank you for your detailed responses and the work you put into running the additional experiments!
> > >
> > > Though lacking of some theoretical analyses (e.g., `empirically much fewer queries are needed`; `the trace stability assumption almost always holds for SAFE queries`, etc.), the new results support your claims, and should be added to the main paper.
> > >
> > > The responses have addressed the concerns and questions I had regarding the paper, thus, I will increase my final score accordingly.

---

> > > > ### Author Response · Authors · 2025-11-20
> > > >
> > > > Thank you very much for taking the time to review our additional experiments and clarifications. We truly appreciate your constructive feedback and are glad that our responses addressed your concerns. We also sincerely thank you for increasing your score. As suggested, we have incorporated the new results into the paper. Thank you again for your careful evaluation and engagement.

---

### Author Response · Authors · 2025-11-30
**Final Comment to the Area Chair**

Dear Area Chair,

Following the rollback of reviewer scores, we would like to provide a clear and complete picture of the state of the reviews prior to the rollback and the status of the discussion at its close.
Initially, the paper was scored (4, 4, 6). During the discussion, reviewer upVP - who had given 4 with confidence 5 - explicitly wrote that our responses had addressed their concerns and that they would increase their final score accordingly, and indeed updated their rating to 6. Reviewer KfRp, who had given 6, followed up to say that they “maintain [their] positive evaluation and score.” The third reviewer, 6kQJ, did not post a follow-up comment before the discussion closed, despite our detailed updates and requests for clarification, so their score remained 4. At the time the scores were reset, the paper therefore stood at **(6, 4, 6)** with confidences **(5, 4, 4)**.

As the remaining reviewer (6kQJ) did not respond during the discussion, we briefly summarize how their concerns were addressed. Their two main points concerned novelty and the handling of UNKNOWN outcomes. In the rebuttal, we clarified, beyond being the first formal verification work on early-exit NNs, there are conceptual challenges unique to early-exit networks—including the need to redefine robustness under conditional execution, the careful treatment of spurious SAFE/UNSAFE results from different exit possibilities, a sound optimization tailored to local robustness with early exits, and our theoretical and empirical justification via the trace-stability assumption. We also highlighted that our work is the first to leverage early exits to accelerate formal verification, yielding scalable and substantial practical improvements. **These are all novel contributions of our work.** Regarding UNKNOWN outcomes, we explained that these stem from solver limits. Since having UNKNOWN/timeout instances is in fact very common in all previous works on NN verification (Brix et al., 2023; 2024), this **should not be viewed as a weakness of our work**. Additionally, verifying smaller early exit sub-networks naturally reduces such instances. Although the reviewer did not follow up, we believe these clarifications **fully addressed** their concerns.

Across all reviews, the strengths of the work were consistently highlighted, including the clear formalization of robustness for early-exit networks, the soundness and completeness guarantees for our algorithms, and the strong empirical evidence showing that early exits substantially improve verifiability.

Several concerns raised by the reviewers could be addressed directly through additional experiments, and we made every effort to run them. We reran the key verification queries on an NVIDIA A100 - using configurations validated by the αβ-CROWN maintainers - and showed that the trends observed on the Apple M3 hold consistently across platforms. We also added a detailed sensitivity study for small variations in (T), further validating the results and demonstrating the importance of having a sound and complete formulation. The remaining points were addressed through complete and careful explanations, including the role of trace stability in different scenarios, the behavior of other verifiers such as Marabou, and the interpretation of UNKNOWN outcomes. We provided thorough and detailed answers, supported by both analysis and evidence, to ensure that all concerns were resolved. Finally, we updated and extended the manuscript to incorporate the conclusions and insights gained during the rebuttal.

Overall, the reviewers who engaged with our responses indicated that their concerns were fully addressed and updated their scores accordingly. Given this progression, and the fact that the remaining reviewer’s concerns were similarly addressed in the rebuttal, we believe the strengthened results and clarifications place the paper in a clearer and more compelling position for final assessment.

---

### Meta-Review · Area_Chair_HK7c · 2026-01-09

**Summary:**

Reviewers are concerned about
- The experimental concerns regarding the hardware has been adequately addressed.
- The role of trace stability in different scenarios and the behavior of other verifiers have been properly addressed in the rebuttal.
- The method only improves runtime of solvable cases.
- The proposed algorithms do not handle UNKNOWN results from the underlying verifier, which appear frequently in the experimental evaluation.
- The proposed algorithms do not handle UNKNOWN results from the underlying verifier, which appear frequently in the experimental evaluation.

**Reviewer Concerns:**

- The experimental concerns regarding the hardware has been adequately addressed.
- The role of trace stability in different scenarios and the behavior of other verifiers have been properly addressed in the rebuttal.

**Reviewer Scores:**

I am not sure if the reviewers would have increased their scores.

---

### Decision · Program_Chairs · 2026-01-26

Reject